# Transgenerational dynamics of rDNA copy number in *Drosophila* male germline stem cells

**Kevin L Lu**[1,2,3†], **Jonathan O Nelson**[1,4†], **George J Watase**[1,4], **Natalie Warsinger-Pepe**[1,5], **Yukiko M Yamashita**[1,2,4,6]*

[1]Life Sciences Institute, University of Michigan, Ann Arbor, United States; [2]Cellular and Molecular Biology Program, University of Michigan, Ann Arbor, United States; [3]Medical Scientist Training Program, University of Michigan, Ann Arbor, United States; [4]Howard Hughes Medical Institute, University of Michigan, Ann Arbor, United States; [5]Department of Molecular and Integrative Physiology, University of Michigan, Ann Arbor, United States; [6]Department of Cell and Developmental Biology, University of Michigan, Ann Arbor, United States

**\*For correspondence:**
yukikomy@umich.edu

[†]These authors contributed equally to this work

**Abstract** rDNA loci, composed of hundreds of tandemly duplicated arrays of rRNA genes, are known to be among the most unstable genetic elements due to their repetitive nature. rDNA instability underlies aging (replicative senescence) in yeast cells, however, its contribution to the aging of multicellular organisms is poorly understood. In this study, we investigate the dynamics of rDNA loci during aging in the *Drosophila* male germline stem cell (GSC) lineage, and show that rDNA copy number decreases during aging. Our study further reveals that this age-dependent decrease in rDNA copy number is heritable from generation to generation, yet GSCs in young animals that inherited reduced rDNA copy number are capable of recovering normal rDNA copy number. Based on these findings, we propose that rDNA loci are dynamic genetic elements, where rDNA copy number changes dynamically yet is maintained through a recovery mechanism in the germline.

DOI: https://doi.org/10.7554/eLife.32421.001

## Introduction

The ribosomal DNA (rDNA) loci consist of tandem repetitive arrays of the rRNA genes, which code for the mature RNA components of ribosomes, flanked by intergenic spacer sequences (IGS) (*Figure 1A*). rDNA loci are considered to be one of the most unstable regions of the genome due to their repetitive nature and high transcriptional activity. First, as an inherent characteristic of repetitive DNA, rDNA can undergo intrachromatid recombination leading to copy number loss and generation of circularized repeat units (extrachromosomal rDNA circles (ERCs)) (*Figure 1A*) (*Sinclair and Guarente, 1997*). Second, the rDNA is highly transcribed even during S phase, leading to possible collisions between replication and transcription machineries. This collision can result in double strand breaks, further contributing to genomic instability of the rDNA (*Helmrich et al., 2013*; *Takeuchi et al., 2003*).

Consistent with the notion that rDNA copy number is unstable, it is widely known that rDNA copy number varies considerably among individuals within a population. For example, it has been shown that *Drosophila melanogaster* exhibits 6-fold range of rDNA copy number (*Lyckegaard and Clark, 1989*; *Lyckegaard and Clark, 1991*), and another study estimated copy number range to be 80–600, where individuals with >130 copies are asymptomatic (*Mohan and Ritossa, 1970*). Similarly, rDNA copy number variation has been observed in many species including budding and fission

**Figure 1.** *Drosophila* male GSCs exhibit perturbations in nucleolar morphology with age. (**A**) Illustration of rDNA destabilization through intrachromatid recombination. (**B**) Apical tip of the testes stained for Fibrillarin (red, nucleolus), Vasa (green, germ cells), Fas III (white, hub) and DAPI (blue). The hub, a major component of the GSC niche, is denoted by the asterisk. GSCs with representative nucleolar morphologies are outlined. Bar: 5 µm. (**C**) Distribution of GSC nucleolar morphology during aging, as a percentage of total GSCs scored (n, number of GSCs scored). P-values from chi-squared test (see methods) is shown.

DOI: https://doi.org/10.7554/eLife.32421.002

yeasts (*James et al., 2009*; *Maleszka and Clark-Walker, 1993*; *Pasero and Marilley, 1993*), planktonic crustaceans *Daphnia* (*Eagle and Crease, 2012*), plants (barley [*Zhang et al., 1990*], bell bean [*Rogers and Bendich, 1987*]), mouse (*Gibbons et al., 2015*) and humans (*Gibbons et al., 2015*; *Gibbons et al., 2014*). Despite striking copy number variation among individuals, the range of copy number is well maintained within the population, suggestive of a mechanism that maintains rDNA copy number through generations. However, underlying molecular mechanisms responsible for maintenance of rDNA copy number are poorly understood.

In yeast, rDNA instability (i.e. reduction of rDNA copy number and associated accumulation of ERCs) is a major cause of replicative aging/senescence (*Ganley and Kobayashi, 2014*; *Kobayashi, 2011*; *Sinclair and Guarente, 1997*). The state of rDNA stability in various mutants that either stabilize or decrease rDNA copy number correlates well with lengthened or shortened life span, respectively (*Ganley and Kobayashi, 2014*). Despite clear involvement of the rDNA in the replicative lifespan of yeast, whether and how rDNA instability contributes to aging of multicellular organisms remains unclear. In particular, stem cells proliferate throughout the life of multicellular organisms, and their declining number and/or function during aging is proposed to be an underlying cause of organismal aging, due to inability to replenish essential cell populations (*López-Otín et al., 2013*).

Despite the clear relationship between rDNA and aging, and that between aging and stem cells, little is known about whether or how the rDNA may change in stem cells during organismal aging.

In this study, we investigate the dynamics of the rDNA loci during aging in *Drosophila* male germline stem cells (GSCs). *Drosophila* male GSCs serve as an excellent experimental paradigm to study rDNA stability during aging, by providing a genetically tractable system to examine the aging of stem cells at a single cell resolution. GSCs undergo continuous asymmetric divisions throughout adulthood, producing one self-renewed GSC and one differentiating cell (*Inaba and Yamashita, 2012*). Additionally, *Drosophila* rDNA loci are limited to the sex chromosomes (X and Y), providing a simplified system to assess rDNA stability compared to other animal models that have multiple rDNA loci spread across many chromosomes (*McStay, 2016*). We show that male GSCs undergo destabilization of rDNA during aging. This rDNA destabilization manifests cytologically as atypical morphology of the nucleolus, the site of rDNA transcription (*Boisvert et al., 2007*; *Pederson, 2011*). We find that rDNA transcription is normally restricted to the Y chromosome in GSCs, as has been observed in other male *Drosophila* cell types (*Greil and Ahmad, 2012*; *Zhou et al., 2012*), but GSCs with atypical nucleolar morphology exhibit transcriptional activation of the normally-silent X rDNA locus. Our results indicate that X rDNA activation is likely to compensate for reduction in rDNA copy number that progresses during aging. We further show that such destabilization of rDNA is heritable, impacting the rDNA copy number of the germline and GSC nucleolar morphology in the next generation. Strikingly, we found that GSCs in the $F_1$ generation are capable of recovering rDNA copy number in the early ages of adulthood, revealing the likely presence of a mechanism that maintains rDNA copy number through generations. We further show that this recovery requires the same factors needed for the phenomenon known as 'rDNA magnification', where rDNA copy number increases in the male germline in the animals with large rDNA deletions (*Hawley and Tartof, 1983*; *Hawley and Tartof, 1985*; *Ritossa, 1968*; *Tartof, 1974*). Taken together, we propose that the rDNA represents dynamic genetic loci that undergo degeneration and recovery during the aging of individuals and through generations. We further propose that rDNA copy number reduction during aging of parents and its recovery in the subsequent generation may contribute to widely observed copy number variation among individuals within species, which is nonetheless maintained within a certain range.

## Results

### Nucleolar morphology is perturbed during the aging of *Drosophila* male germline stem cells

To investigate the potential destabilization of rDNA in *Drosophila* male GSCs, we first examined nucleolar morphology during aging. The nucleolus is organized by transcription of the rDNA, thus its morphology is expected to reflect the transcriptional activity of rDNA loci (*Boisvert et al., 2007*; *Pederson, 2011*). Immunofluorescence staining of the nucleolar component Fibrillarin (*Ochs et al., 1985*) showed that most GSCs in young males contain a single, round nucleolus approximately 2 µm in diameter (89.2%, n = 408, *Figure 1B*, 'typical'). However, we found that the frequency of GSCs with typical nucleolar morphology progressively decreased during aging throughout 40 days of adulthood (*Figure 1C*), as the GSCs with atypical nucleolar morphology increased. Atypical nucleolar morphology can be categorized into two types: (1) fragmented, where more than one distinct nucleolar foci were observed, and (2) deformed, where the nucleolus lost its typical compact, round morphology (*Figure 1B*, 'fragmented' and 'deformed'). Although fragmented and deformed nucleoli are sometimes difficult to distinguish from each other and may indeed represent the same population of cells, we scored them separately to protect against losing information. However, these two populations behaved similarly in most assays performed in this study. These results show that nucleolar morphology is progressively perturbed during the aging of *Drosophila* male GSCs, prompting us to further investigate the underlying causes.

### Perturbed nucleolar morphology is associated with transcriptional activation of the normally silent X chromosome rDNA locus

To investigate the underlying cause(s) of perturbed nucleolar morphology, we first examined the spatial relationship between the nucleolus and the rDNA loci that organize nucleolus formation. In

*Drosophila melanogaster*, the rDNA loci are located on the sex chromosomes (X and Y), each containing ~200–250 copies of rDNA (*Figure 2A*) (*Ritossa et al., 1971*). The X and Y rDNA loci can be detected by DNA fluorescence *in situ* hybridization (FISH) using chromosome-specific probes against the $(AATAAAC)_n$ satellite and the 359 bp repeat satellite, which are adjacent to the Y and X rDNA loci, respectively (*Figure 2A*). By combining DNA FISH with immunofluorescence staining to detect nucleoli, we found that the Y rDNA locus is always associated with the nucleolus when GSCs have typical nucleolar morphology, whereas the X rDNA locus was not, irrespective of age (*Figure 2B*). Because the assembly of the nucleolus reflects rDNA transcription, these results suggest that Y rDNA is actively transcribed, whereas X rDNA is not. This likely reflects a phenomenon known

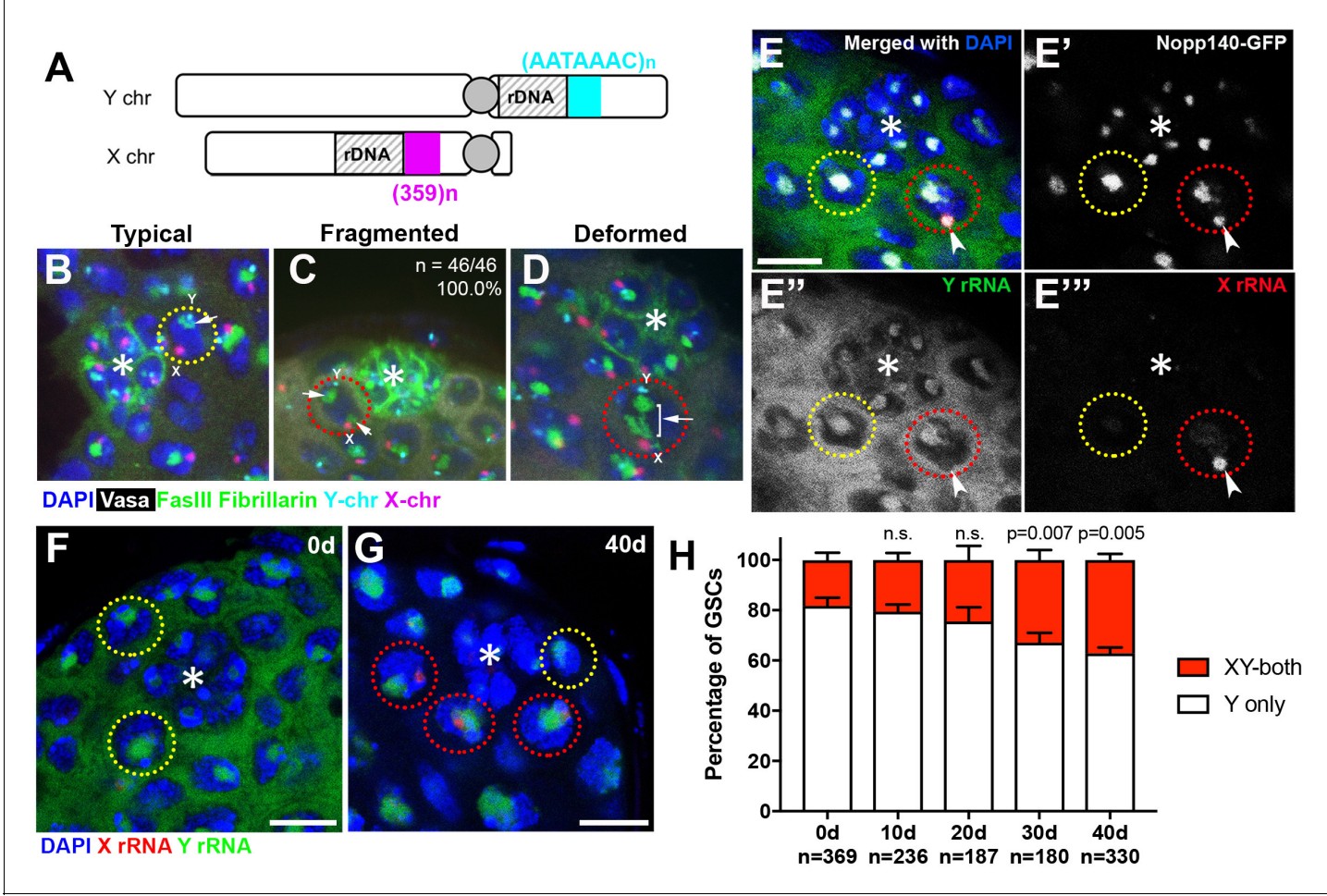

**Figure 2.** Transcriptional activation of X rDNA in GSCs with atypical nucleolar morphology. (**A**) Illustration of rDNA loci on *Drosophila* X and Y chromosomes. Y rDNA locus is juxtaposed to $(AATAAAC)_n$ satellite repeats (cyan), whereas X rDNA locus is juxtaposed to 359 bp satellite repeats (magenta). (**B–D**) DNA FISH for the 359 bp satellite repeats (magenta) and $(AATAAAC)_n$ satellite repeats (cyan), combined with immunofluorescence staining for Vasa (white), FasIII/Fibrillarin (green), DAPI (blue). B: typical nucleolus, C: fragmented nucleoli, D: deformed nucleolus. The hub is denoted by (*). GSCs with typical nucleoli are indicated by yellow dotted lines, GSCs with atypical nucleoli are indicated by red dotted lines. Arrows indicate the position of nucleoli. (**E**) SNP *in situ* hybridization with Y and X chromosome-specific rRNA probes combined with Nopp140-GFP to mark nucleolar morphology. Y rRNA (green), X rRNA (red), Nopp140-GFP (white), DAPI (blue). Bar: 7.5 μm. (**F, G**) SNP *in situ* hybridization in the testes from 0 day (**F**) and 40 day (**G**) old flies. GSCs with only Y rRNA (yellow outline) and with both X and Y rRNA transcription (red outline). Y rRNA (green), X rRNA (red), DAPI (blue). (**H**) XY rRNA transcription during aging of GSCs, as a percentage of total GSCs scored (n, number of GSCs scored). Mean ±SD (p-value of t-test is indicated). Note that 'X-only' rRNA transcription was never observed.

DOI: https://doi.org/10.7554/eLife.32421.003

The following figure supplement is available for figure 2:

**Figure supplement 1.** SNP-FISH is highly specific for rRNA transcribed from the Y vs X chromosomes.
DOI: https://doi.org/10.7554/eLife.32421.004

as 'nucleolar dominance', where only certain rDNA loci (Y rDNA in the case of *D. melanogaster* males) are actively transcribed while the others (X rDNA in *D. melanogaster* males) are silent (see below) (*Greil and Ahmad, 2012*; *Zhou et al., 2012*).

Interestingly, when the nucleolus is fragmented, the ectopic nucleolus (typically the smaller one) was almost always closely located near the X rDNA locus, irrespective of age (*Figure 2C*, n = 46/46, 100.0% in 0–1 day old flies, n = 46/47, 97.9% in 40 day old flies). X rDNA was also associated with deformed nucleoli (*Figure 2D*). These results suggest that the X chromosome has gained nucleolar organizing activity (i.e. became transcriptionally active) in GSCs when the nucleolar morphology is atypical.

To directly test the idea that atypical nucleolar morphology is associated with transcriptional activation of the normally inactive X rDNA locus, we adapted single nucleotide polymorphism (SNP) RNA *in situ* hybridization to differentially visualize the X and Y rDNA transcripts (*Levesque et al., 2013*) (*Figure 2—figure supplement 1A*). By genetically isolating and sequencing rDNA arrays from the X and Y chromosomes, we identified four SNPs in the coding and ITS sequences between the X and Y rDNA loci of the wild type strain used in this study (*yw*) (see methods). We designed probes ('SNP probes') utilizing these four SNPs to distinguish X- vs. Y-derived rRNA transcripts (*Supplementary file 1*). Specificity of SNP probes was confirmed by SNP *in situ* hybridization in X/O males (containing only X rDNA), and C(1)DX/Y females (containing only Y chromosome rDNA), where only the expected SNP signals were observed (*Figure 2—figure supplement 1B*).

We combined SNP *in situ* hybridization with the nucleolar marker Nopp140-GFP (*McCain et al., 2006*) to correlate X- vs. Y-derived rRNA transcription with nucleolar morphology. We found that most GSCs only transcribed Y rRNA (*Figure 2E,F,H*), demonstrating that nucleolar dominance indeed occurs in the male germline, as observed in male larval neuroblasts (*Greil and Ahmad, 2012*; *Zhou et al., 2012*). In all GSCs with fragmented nucleoli, one nucleolus (typically the larger one) showed a Y SNP signal, whereas the other showed an X SNP signal, supporting the idea that nucleolar fragmentation is associated with transcriptional activation of the X rDNA locus (*Figure 2E,G*, n > 40). As flies age, many GSCs exhibited expression of rRNA from both the Y and X chromosomes (*Figure 2G,H*). The number of GSCs expressing X rRNA increased from 18.2 ± 3.0% at 0 days old to 37.2 ± 2.4% at 40 days old (*Figure 2H*). These results, together with the above result that showed association of the X rDNA locus with fragmented/deformed nucleoli (*Figure 2A–F*), strongly suggest that atypical nucleolar morphology that accumulates in GSCs during aging is due to activation of the normally silent X rDNA locus, causing rDNA transcription from two separate chromosomes, each forming distinct nucleoli.

## Y chromosome rDNA copy number decreases in the male germline during aging

Why does the X rDNA locus activate in aging GSCs? We hypothesized that rDNA copy number might be reduced during aging due to the inherent instability of the repetitive locus, requiring compensatory activation of a normally silent rDNA locus (i.e. X rDNA) to meet the cellular requirement for rRNA transcription. It has been extensively shown in yeast that stability of the rDNA is compromised during aging, with intrachromatid recombination leading to loss of rDNA on the chromosomes (*Ganley et al., 2009*; *Ganley and Kobayashi, 2014*; *Kobayashi, 2008*), leading us to hypothesize that a similar process may underlie the aging of *Drosophila* male GSCs.

To address the possibility that rDNA copy number might be decreased during aging in male GSCs, we first isolated genomic DNA from testes of young and old flies and quantified their rDNA copy number using a previously published qPCR-based method for quantifying rDNA copy number (*Supplementary file 2*) (*Aldrich and Maggert, 2014*), and found a significant reduction in rDNA copy number (*Figure 3A*). This copy number loss was observed across all the mature rRNA genes in the 45S cistron (18S, 5.8S and 28S rRNA genes). Interestingly, the copy number of R1 and R2 retrotransposable elements did not decrease with age (*Figure 3A*). R1 and R2 retrotransposable elements selectively insert into the 28S rRNA gene, and inserted rDNA units are transcriptionally repressed (*Ye and Eickbush, 2006*). These results suggest that the rDNA copy number loss primarily occurs in uninserted, actively-transcribed rDNA. These results imply that loss of rDNA copy number is associated with its transcriptional activity, consistent with the well-established notion that collision between transcription and replication machineries causes rDNA instability (*Helmrich et al., 2013*; *Takeuchi et al., 2003*).

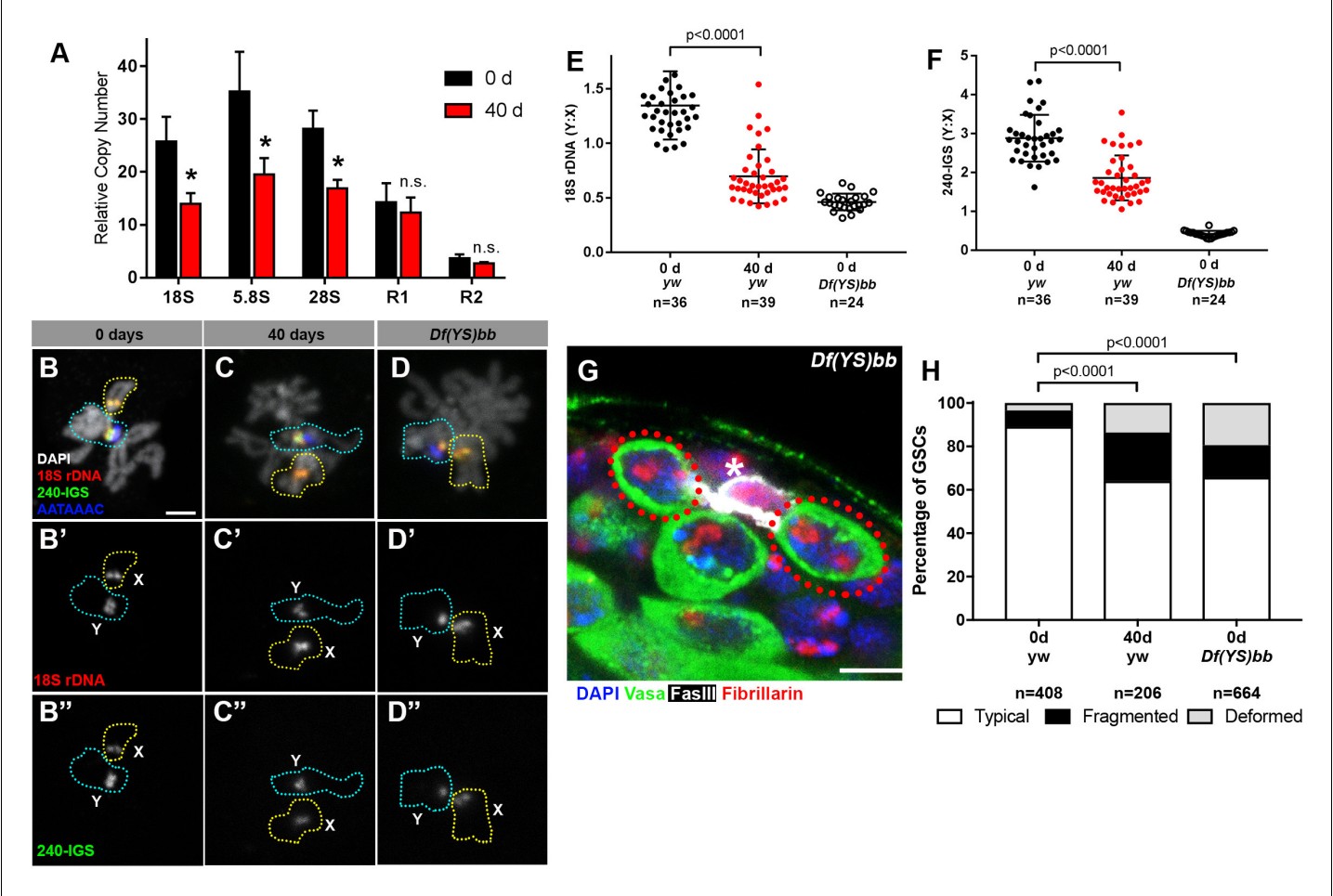

**Figure 3.** rRNA gene copy number decreases in germ cells during aging. (A) rRNA gene copy number quantification by qPCR from 0 day and 40 day old testes. Mean ±SD (p-value *$\leq$0.05 t-test). (B–D) FISH on testis mitotic chromosome spreads. DAPI (white), 18S rDNA (red), 240 bp IGS (green), (AATAAAC)$_n$ (blue). Bar: 2.5 μm. Y chromosome, identified by the presence of (AATAAAC)$_n$, is indicated by cyan outline, and X chromosome is indicated by yellow outline. (E) Y:X signal intensity ratio for the 18S rDNA in mitotic germ cells in day 0, day 40 old wild type (yw) testes, and X/Df(YS)bb testes. Bracket indicates mean ±SD. p-values from Student's t-test is shown. (F) Y:X signal intensity ratio for the IGS in mitotic germ cells in day 0, day 40 old wild type (yw) testes, and X/Df(YS)bb testes. Bracket indicates mean ±SD. p-values from Student's t-test is shown. Note that different Y:X ratios for 18S vs. IGS probes indicates that Y rDNA locus might have higher number of IGS repeats per rDNA unit. (G) Examples of GSCs with atypical nucleolar morphology from X/Df(YS)bb flies at 0 days (red outline). Fibrillarin (red), DAPI (blue), Vasa (green), FasIII (white). The hub is denoted by (*). Bar: 5 μm. (H) Distribution of GSC nucleolar morphologies in X/Df(YS)bb flies compared to 0 and 40-day-old WT flies, as a percentage of total GSCs scored (n, number of GSCs scored). Chi-squared test, p-values listed.

DOI: https://doi.org/10.7554/eLife.32421.005

The following source data and figure supplement are available for figure 3:

**Source data 1.** Pixel intensity measurement and its ratio from DNA FISH plotted in **Figure 3E,F**.

DOI: https://doi.org/10.7554/eLife.32421.007

**Figure supplement 1.** The Df(YS)bb⁻ from Kyoto Stock Center chromosome does not have any detectable 240-IGS sequence.

DOI: https://doi.org/10.7554/eLife.32421.006

Given that Y rDNA is predominantly transcribed in most young GSCs (**Figure 2H**), and that rDNA copies with retrotransposon insertion, which are known to be mostly silenced (**Ye and Eickbush, 2006**), are not lost during aging (**Figure 3A**), we hypothesized that the transcriptionally active Y rDNA copies are more frequently lost than normally silent X rDNA copies during aging. To assess this possibility, we used a quantitative DNA fluorescence *in situ* hybridization (FISH) method to examine changes in relative copy number of rDNA on X and Y chromosomes during aging (see methods). DNA FISH was performed on chromosome spreads from mitotic spermatogonia and

meiotic spermatocytes (at stages when the X and Y chromosomes are not paired) using differentially-labelled 18S rDNA, 240-IGS and (AATAAAC)n probes, where (AATAAAC)n marked Y chromosomes. Then, relative fluorescence intensity of 18S rDNA and 240-IGS signals between X and Y chromosomes were determined (*Figure 3B–F*). By using this method, we found that Y:X ratio of the 18S rRNA gene was $1.35 \pm 0.31$ and that of the 240 bp repeat intergenic spacer (240-IGS) was $2.88 \pm 0.60$ in wild type (*yw*) at day 0, indicating that Y chromosome harbors slightly more rDNA than X chromosome in the wild type strain (*yw*) used in this study. To confirm the sensitivity of this quantitative DNA FISH method, we conducted DNA FISH on the chromosomes from animals with large deletions of Y rDNA. First, when DNA FISH was performed using a strain harboring a Y chromosome with a near complete loss of rDNA (*Df(YS)bb⁻*, obtained from Kyoto Stock Center, described in [*Endow, 1982*]) (*Figure 3—figure supplement 1A–B*), no signal was detected on Y chromosomes, suggesting that our FISH method has a very low background. Second, we were able to detect a reduction in rDNA of another Y chromosome harboring an rDNA deletion (*Df(YS)bb*, obtained from Bloomington Stock Center, described by *Cline [2001]*). When this Y chromosome was combined with the X chromosome of the wild type strain (*yw*), we detected that *Df(YS)bb*:X ratio of the 18S rDNA was reduced to $0.46 \pm 0.08$ from $1.35 \pm 0.31$ of the wild type (*Figure 3E*), and *Df(YS)bb*:X 240-IGS ratio was dropped to $0.42 \pm 0.08$ from $2.88 \pm 0.60$ of the wild type (*Figure 3F*), demonstrating that this method can detect partial deletion of rDNA. These results suggest that our DNA FISH method is sensitive enough to distinguish differences in the relative copy number of X and Y chromosome rDNA loci between different conditions, although it might not be fully quantitative.

By using this method, we compared Y:X rDNA ratio in day 0 vs. day 40 old testes. We found that the Y:X 18S rRNA gene ratio dropped from $1.35 \pm 0.31$ to $0.70 \pm 0.25$ by 40 days (*Figure 3E*) and that the Y:X 240-IGS ratio reduced from $2.88 \pm 0.60$ at 0 days to $1.86 \pm 0.58$ by 40 days (*Figure 3F*). Although the quantitative FISH method is only capable of detecting the ratio between X and Y, but not the absolute amount on each chromosome, the fact that overall germline rDNA copy number decreases during aging as demonstrated by qPCR (*Figure 3A*) suggests the change in Y:X ratio reflects Y rDNA loss, instead of X rDNA expansion. It should be noted that these data do not exclude the possibility that rDNA copies are also lost from the X chromosome. However, decreased Y:X ratio during aging suggests preferential loss of Y rDNA.

Since all adult germ cells are derived from GSCs, the loss of Y rDNA copies in the germline (detected by qPCR and quantitative FISH) suggests that Y rDNA copy number is reduced in GSCs. The loss of Y chromosome rDNA may lead to compensatory activation of X rDNA and the atypical nucleolar morphology observed during aging in GSCs. To directly address the causal relationship whether reduced rDNA copy number on the Y chromosome causes disrupted nucleolar morphology in GSCs, we examined GSC nucleolar morphology in X/*Df(YS)bb* flies, which harbors a partial deletion of Y rDNA. Even at a young age, GSCs in these flies exhibited atypical nucleolar morphology at a frequency comparable to 40-day-old wild type flies (*Figure 3G,H*). Taken together, we propose that rDNA copy number decreases during aging in male GSCs, which more profoundly occurs to the Y chromosome rDNA likely due to its transcriptionally active state. The reduction in rDNA copy number on the Y chromosome then leads to compensatory activation of X rDNA in GSCs, causing atypical nucleolar morphology.

## GSC nucleolar morphology and rDNA loss is heritable

Because GSCs are the progenitors of gametes, we reasoned that rDNA copy number loss during the aging of the germline may be heritable to the next generation. To test this idea, we examined the $F_1$ sons from old (day 40) parents ($P_0$) (*Figure 4A*). qPCR on genomic DNA collected from testes of newly-eclosed $F_1$ flies showed a significant reduction in the rDNA copy number when compared to young $P_0$ flies (0d $P_0$), but similar to 40 day old $P_0$ (40d $P_0$) (*Figure 4B*). Again, the number of R1- and R2-inserted rDNA copies was not significantly affected. These results suggest that old parents transmit reduced rDNA copy number to their offspring.

Consistent with the reduced copy number of rDNA in $F_1$ sons measured by qPCR, nucleolar morphology in $F_1$ GSCs from old parents was also perturbed, with only 59.1% of GSCs displaying typical nucleolar morphology in 0-day-old $F_1$ sons (0d $F_1$), similar to 40-day-old $P_0$ fathers (40d $P_0$) (*Figure 4C*). Furthermore, SNP *in situ* hybridization demonstrated similar levels of activation of the X rDNA locus in 40-day-old $P_0$ and 0-day-old $F_1$ GSCs from the sons of old fathers, consistent with disrupted Y rDNA copy number in these flies (*Figure 4D*). Reduction in Y rDNA copy number in $F_1$ sons

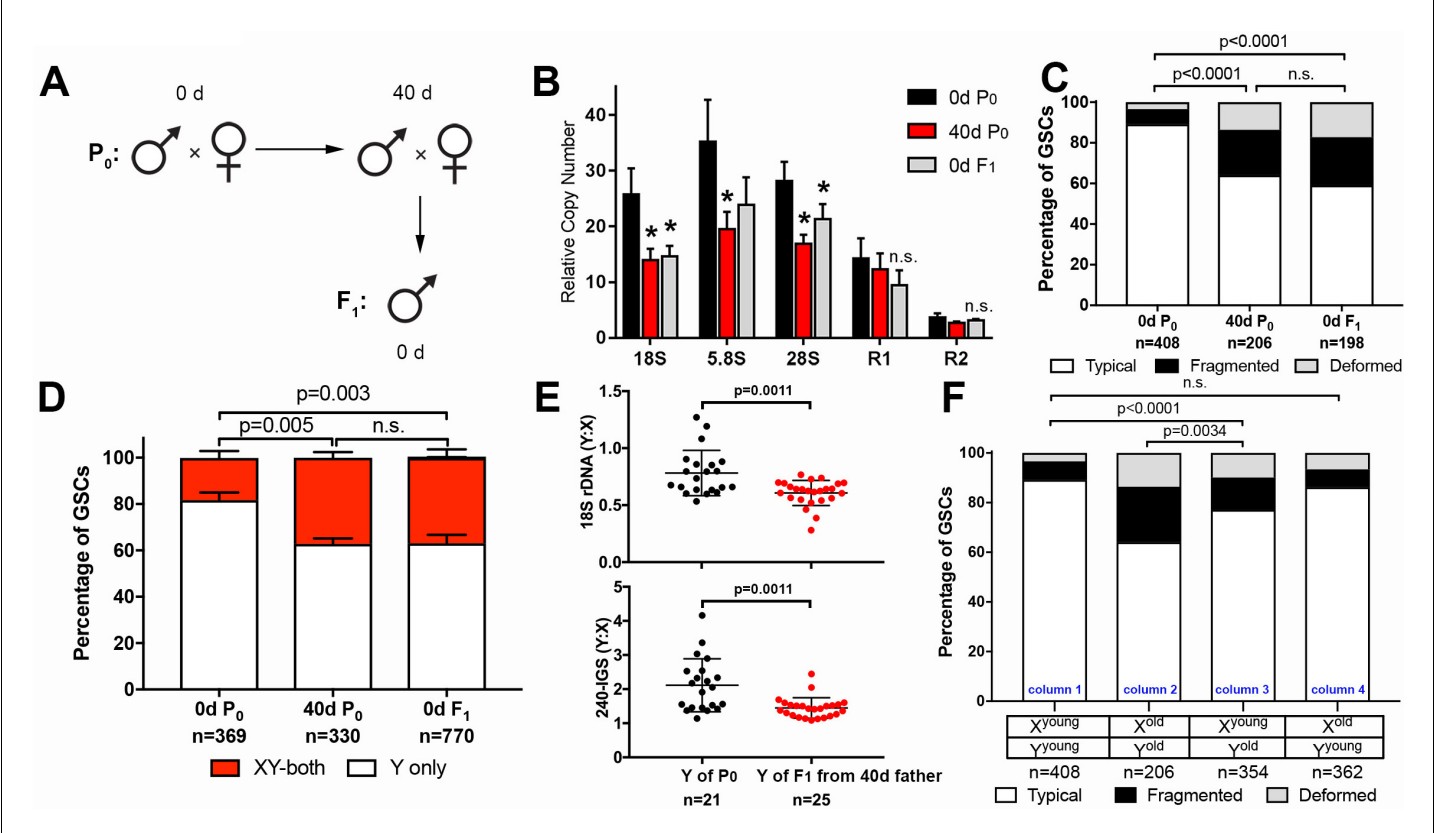

**Figure 4.** GSC nucleolar morphology and rDNA copy number decrease is heritable. (A) Scheme for aging of flies and collection of $F_1$ progeny from old parents. (B) rDNA quantification from testes by qPCR in $P_0$ at 0 and 40 days, and $F_1$ at 0 days. Mean ±SD (p-value *$\leq$0.05, t-test). (C) GSC nucleolar morphology in young $P_0$, old $P_0$ and young $F_1$, as a percentage of total GSCs scored (n, number of GSCs scored). p-values from chi-squared test are shown. (D) Nucleolar dominance assessed by SNP *in situ* in GSCs from young $P_0$, old $P_0$ and young $F_1$ (n, number of GSCs scored). Mean ±SD. p-value of t-test is shown. Note that 'X-only' rRNA transcription was never observed. (E) Y:X signal intensity ratio for the 18S rDNA and 240-IGS in mitotic germ cells comparing day 0 $P_0$ Y and day 0 $F_1$ Y (from day 40 father). Day 0 vs. day 40 fathers ($P_0$) were mated to day 0 old females to yield $P_0$ Y/X vs. $F_1$ Y/X ratio, where X comes from the same source (day 0 yw female). Bracket indicates mean ±SD. p-values from Student's t-test is shown. (F) Effect of X and Y chromosome inheritance from young vs. old parents on nucleolar morphology, as a percentage of total GSCs scored (n, number of GSCs scored). P-value from chi-squared test is shown.

DOI: https://doi.org/10.7554/eLife.32421.008

The following source data is available for figure 4:

**Source data 1.** Pixel intensity measurement and its ratio from DNA FISH plotted in *Figure 4E*.
DOI: https://doi.org/10.7554/eLife.32421.009

was further confirmed by DNA FISH (*Figure 4E*): 0 day vs. 40-day-old $P_0$ (father) was mated to young female and Y: X ratio of 18S and 240-IGS was determined in young (day 0) $F_1$ sons. Because X chromosomes were inherited from young females in both cases, the Y: X ratio allowed direct comparison of Y rDNA in $P_0$ vs. $F_1$. These data clearly suggest reduction in rDNA copy number on the Y chromosome of $F_1$ sons from old fathers.

Examining GSC nucleolar morphology in progeny from multiple genetic crosses provided insight into the dynamics of rDNA copy number on X and Y chromosomes in aged fathers. When 40 day old fathers (contributing $Y^{old}$ to their sons) were crossed to young mothers (contributing $X^{young}$ to their sons), the newly-eclosed $F_1$ sons ($X^{young}/Y^{old}$) displayed disrupted GSC nucleolar morphology, suggesting that Y chromosomes from aged fathers are compromised (*Figure 4F*, column three compared to column 1). In contrast, when young fathers (contributing $Y^{young}$) were crossed to 40 day old virgin mothers (contributing $X^{old}$), their offspring containing $X^{old}$ and $Y^{young}$ chromosomes did not show perturbed nucleolar morphology (*Figure 4F*, column one vs. column 4), suggesting that male GSC nucleolar morphology is determined solely by the age of the inherited Y chromosome. This

may be in part due to the dominance of the Y chromosome rDNA locus over the X rDNA locus, and does not necessarily reveal the state of the maternally inherited X chromosome, or whether the X chromosome undergoes degeneration during aging in females. However, the state of the X chromosome ($X^{young}$ vs. $X^{old}$) passed from the mother appears to affect the degree of disruption of nucleolar morphology in the context of the Y chromosome from the old father ($Y^{old}$) (*Figure 4F*, column 2 and column 3). This might suggest that the X chromosome rDNA might also undergo degeneration in the female germline during aging. Taken together, we conclude that rDNA copy number reduction is heritable and passed to the offspring from old fathers.

## Germline rDNA recovers in the $F_1$ generation

Although $F_1$ sons from old fathers started with disrupted GSC nucleolar morphology, we unexpectedly found that nucleolar morphology recovered as these $F_1$ sons age (*Figure 5A*). Interestingly, the recovery was specifically observed during the first 10 days after eclosion. At 10 days, $F_1$ sons from old fathers recovered to the point where they were comparable to $F_1$ sons of the same age from young fathers, after which nucleolar morphology in these two populations worsened at a similar rate (*Figure 5A*, compare to *Figure 1D*). Concomitant with the recovery of nucleolar morphology, X rDNA expression in $F_1$ sons from old fathers diminished until 10 days, after which it again increased (*Figure 5B,C*). During this period, Y: X rDNA ratio also recovered (*Figure 5D*), revealing remarkable ability of the Y chromosome rDNA to expand in copy number. Together, the recovery of GSC nucleolar morphology and X rDNA repression, combined with the increase in Y:X rDNA ratio during this recovery period in germ cells, suggests that rDNA copies are expanded in GSCs to restore normal state of rDNA transcription (i.e. Y dominant). Interestingly, while we found many germ cells to have recovered Y:X rDNA ratio, there was also a large number of germ cells that apparently did not recover (*Figure 5D*), suggesting that rDNA expansion may be a stochastic event that occurs in individual GSCs.

## Recovery of GSC nucleolar morphology depends on the homologous recombination repair pathway

The observed recovery of GSC rDNA in the sons of aged fathers resembles the phenomenon called 'rDNA magnification.' Animals with rDNA insufficiency due to large deletions of rDNA develop a cuticular and bristle length defect called the *bobbed* phenotype (*Ritossa et al., 1966*). rDNA magnification is the observation that this defect is reverted to a wild type cuticle and bristle in a small subset of offspring from *bobbed* animals, due to the rapid expansion of rDNA copies (*de Cicco and Glover, 1983*; *Ritossa, 1968*). Although the molecular mechanisms of rDNA magnification are not fully understood (*Bianciardi et al., 2012*; *Paredes and Maggert, 2009*; *Ritossa, 1968*; *Robbins, 1996*), it has been shown that the genes involved in the homologous recombination repair are required for rDNA magnification. Specifically, mutations in *mus-101* (*Drosophila* homolog of TOPBP1 (DNA topoisomerase 2-binding protein 1)) and *mei-41* (*Drosophila* homolog of ATR), factors necessary for the resolution step of the homologous recombination-mediated repair of DNA double-strand breaks have severely reduced rates of rDNA magnification (*Hawley and Tartof, 1983*; *Hawley and Tartof, 1985*). Given the resemblance between rDNA magnification from large rDNA deletion and rDNA recovery in young sons from old fathers, we hypothesized that the same mechanism might underlie these phenomena. Indeed, we found that *mus-101* and *mei-41* mutants failed to recover typical nucleolar morphology in the GSCs of animals that inherited their Y chromosome from aged fathers (*Figure 6A–C*, compare to *Figure 5A*). Conversely, mutations in *mus-102*, which is required for DNA damage repair but not for rDNA magnification (*Hawley and Tartof, 1985*), had no effect on the recovery of GSC nucleolar morphology (*Figure 6D*). These results indicate that similar mechanisms might underlie these two phenomena, and that rDNA magnification may be the manifestation of the mechanisms that normally maintains rDNA copy number.

## *mus-101* is required for normal germline rDNA maintenance during aging

Since we found that *mei-41* and *mus-101* are required for the recovery of GSC nucleolar morphology in $F_1$ sons from old fathers, we wondered if these factors also contribute to the maintenance of rDNA during aging. However, wild type control (*yw*), *mei-41*, and *mus-101* mutants exhibited

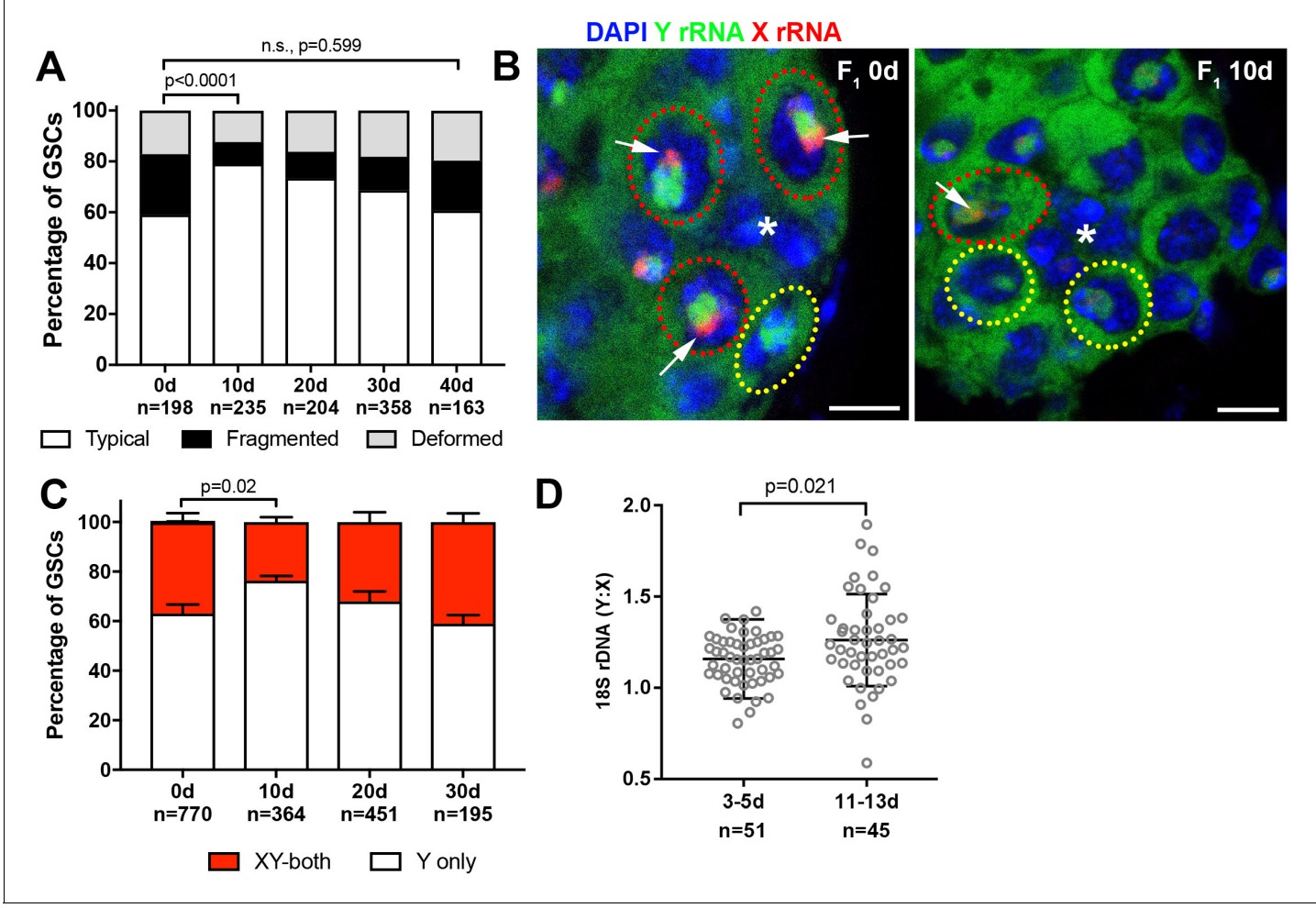

**Figure 5.** Recovery of GSC nucleolar morphology, Y rDNA dominance and Y rDNA copy number in $F_1$ flies. (**A**) Changes in GSC nucleolar morphology in $F_1$ flies from old parents, as a percentage of total GSCs scored (n, number of GSCs scored). p-value from chi-squared test is shown. (**B**) Nucleolar dominance in GSCs from day 0 and day 10 old $F_1$ testes assessed by SNP *in situ* hybridization. DAPI (blue), Y rRNA (green), X rRNA (red). The hub is denoted by (*). Bars: 5 μm. Co-dominant GSCs are indicated by red dotted lines, Y-dominant GSCs are indicated by yellow dotted lines. Arrows indicate X rRNA signal, thus co-dominance. (**C**) Nucleolar dominance in $F_1$ GSCs during aging, as a percentage of total GSCs scored (n, number of GSCs scored). Mean ±SD. P-value of t-test is shown. Note that X-only rRNA transcription was never observed, except for once (out of 770 cells) at day 0, which is included in the graph. (**D**) Ratio of Y:X signal intensity for the 18S rDNA from mitotic chromosome spread of germ cells in $F_1$ flies. Mean ±SD, t-test.

DOI: https://doi.org/10.7554/eLife.32421.010

The following source data is available for figure 5:

**Source data 1.** Pixel intensity measurement and its ratio from DNA FISH plotted in *Figure 5D*.
DOI: https://doi.org/10.7554/eLife.32421.011

unequal rates of atypical nucleolar morphology in GSCs at day 0, likely due to background differences in baseline X rDNA copy number (*Figure 6B–C*, first columns). This variation in initial nucleolar morphology made it difficult to directly compare changes in GSC rDNA content during aging between these genotypes. To control these background differences in rDNA copy number, we crossed a standard Y chromosome from our wild type strain (*yw*) into wild type X ($X^{yw}$) or *mus-101$^{D1}$* mutant background (note that *mus-101* is an X-linked gene) (*Figure 7A,F1*). Then these Y chromosomes were allowed to age for 40 days in their respective genetic backgrounds. These Y chromosomes were isolated by crossing the $F_1$ males to *yw* females, and the state of $F_1$ Y chromosomes were assessed by nucleolar morphology of $F_2$ GSCs (*Figure 7A*). This scheme allowed the comparison of Y chromosomes that have undergone aging in *yw* or *mus-101$^{D1}$* mutant background using the

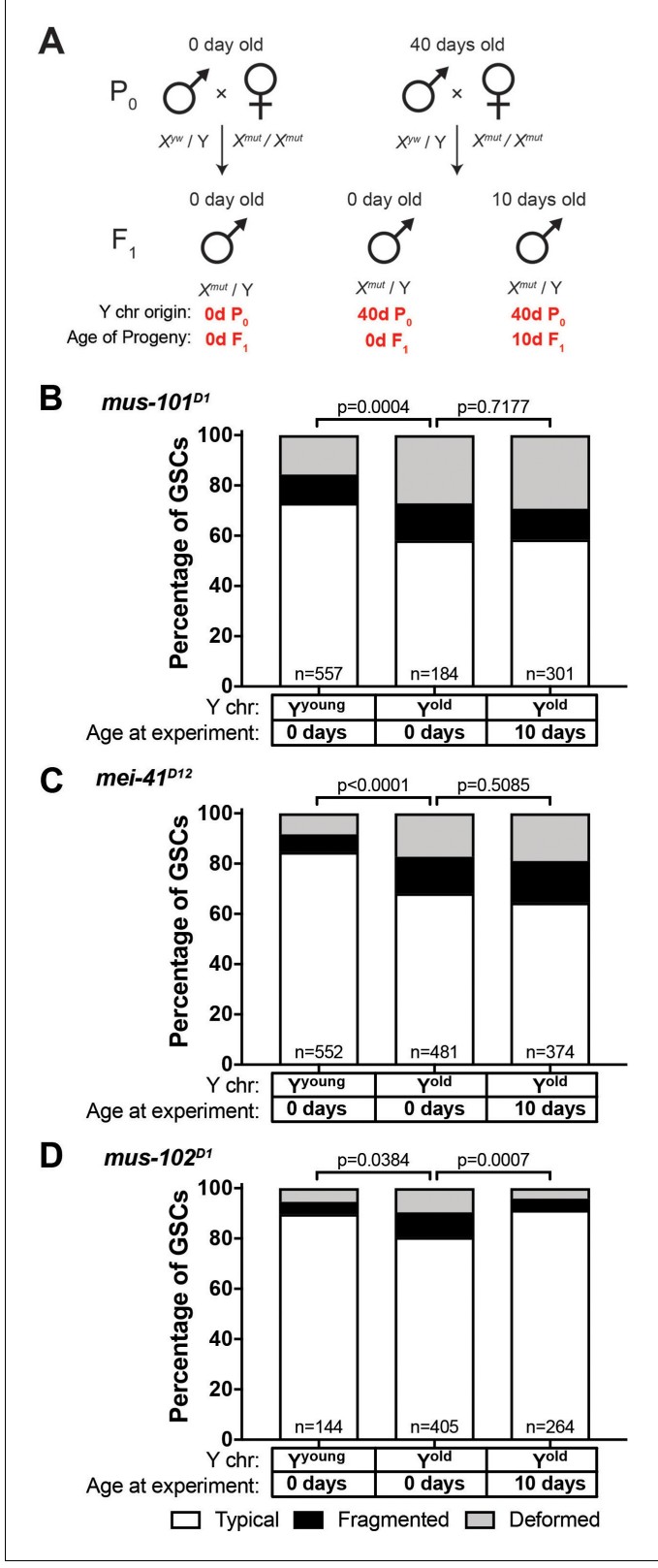

**Figure 6.** Recovery of GSC nucleolar morphology requires *mus-101* and *mei-41*. (**A**) Mating scheme to assess the ability to recover nucleolar morphology after inheriting compromised Y chromosome from old fathers. *yw* males (0 or 40 days old) were mated to 0 day old *mus-101^{D1}*, *mei-41^{D12}*, or *mus-102^{D1}* mutant females. GSC nucleolar morphology in $F_1$ mutant males was examined by anti-Fibrillarin antibody at day 0 or 10. (**B–D**) GSC nucleolar

*Figure 6 continued on next page*

*Figure 6 continued*

morphology in 0 and 10 day-old F$_1$ *mus-101$^{D1}$* (B), *mei-41$^{D12}$* (C), and *mus-102$^{D1}$* (D) mutants, as a percentage of total GSCs scored (n, number of GSCs scored). P-values from chi-squared test between indicated conditions are shown.

DOI: https://doi.org/10.7554/eLife.32421.012

same source of X and Y chromosomes. This scheme effectively eliminated the effects of background variation, as we found that there was no difference in the fraction of GSCs with atypical nucleolar morphology between the sons of 0 day old *yw* and *mus-101$^{D1}$* mutants (**Figure 7B**, n.s (p=0.1647)). However, there was a significant difference between the sons of 40-day-old *yw* and *mus-101$^{D1}$*mutant (**Figure 7B**, p=0.0070), suggesting that the *mus-101* mutant fathers suffer more Y chromosome rDNA loss during aging compared to the *yw* fathers. This result suggests that the same molecular machineries might underlie normal germline rDNA maintenance and the phenomenon of rDNA magnification.

## Discussion

This study provides evidence that rDNA loci are highly unstable but actively maintained genetic loci. Our data shed light onto a few longstanding questions and also raise new questions for future investigation.

### Destabilization of rDNA loci during aging in *Drosophila* male GSCs

Our data show that rDNA copy number decreases during aging of *Drosophila* male GSCs. Although early observations indicated that rDNA content may decrease during aging in mammals (*Johnson and Strehler, 1972*; *Strehler, 1986*), it was observed using bulk tissues containing mostly post-mitotic cells, and its implication in aging of multicellular organisms has been poorly explored. It was shown that in mouse hematopoietic stem cells, replicative stress is a major driver of stem cell aging (*Flach et al., 2014*). Curiously, they observed signs of replication stress (accumulation of γ-H2Ax) mainly in the nucleolus, but it remained unclear why the nucleolus specifically accumulates replication stress. We speculate that destabilization of rDNA loci may underlie age-associated accumulation of replication stress in the nucleolus. No matter how many rDNA copies are present in the genome, approximately, the same number of copies must be transcribed to support the cellular demands for ribosome biogenesis. This constant requirement for rDNA transcription means that those cells with fewer rDNA copies will have a larger proportion of their rDNA being transcribed than cells with more copies, as has been shown in yeast. Replication through actively transcribing rDNA creates the possibility for collision between replication and transcription machineries. Large rDNA arrays may be able to avoid such collision by selectively transcribing rDNA copies that are not undergoing DNA replication at the moment. However, smaller arrays would be limited in their ability to avoid collision due to their requirement to transcribe a larger portion of their rDNA. Increased collisions between replication and transcription machineries has been observed in a yeast strain with reduced rDNA copy number, leading to replication stress (*Takeuchi et al., 2003*). Consistent with the idea that transcription increases the probability of collision and thus destabilization, we observed that the dominantly-transcribed Y chromosome rDNA preferentially underwent destabilization.

Our results revealed a decrease in chromosomal rDNA copy number (predominantly on the Y chromosome) during aging. However, our study does not exclude the potential contribution of ERCs. In yeast, ERCs are specifically segregated to mother cells, whereas daughter cells are devoid of ERCs, potentially explaining the mechanism by which daughter cells reset their age (*Shcheprova et al., 2008*; *Sinclair and Guarente, 1997*). Although detection of ERCs in GSCs (compared to their daughter cells, gonialblasts) at a single cell resolution is not currently possible, it will be of future interest to investigate whether GSCs do accumulate ERCs and, if so, whether their inheritance is asymmetric.

We also found that uninserted copies are selectively lost during aging, as R1 and R2 retrotransposon abundance was maintained. Mathematical modeling by Zhou and Eickbush (*Zhou et al., 2013*) using *D. simulans* X rDNA locus suggested that transcription of rDNA preferentially occurs in contiguous blocks of uninserted copies, and rDNA copy loss preferentially occurs in such blocks. This

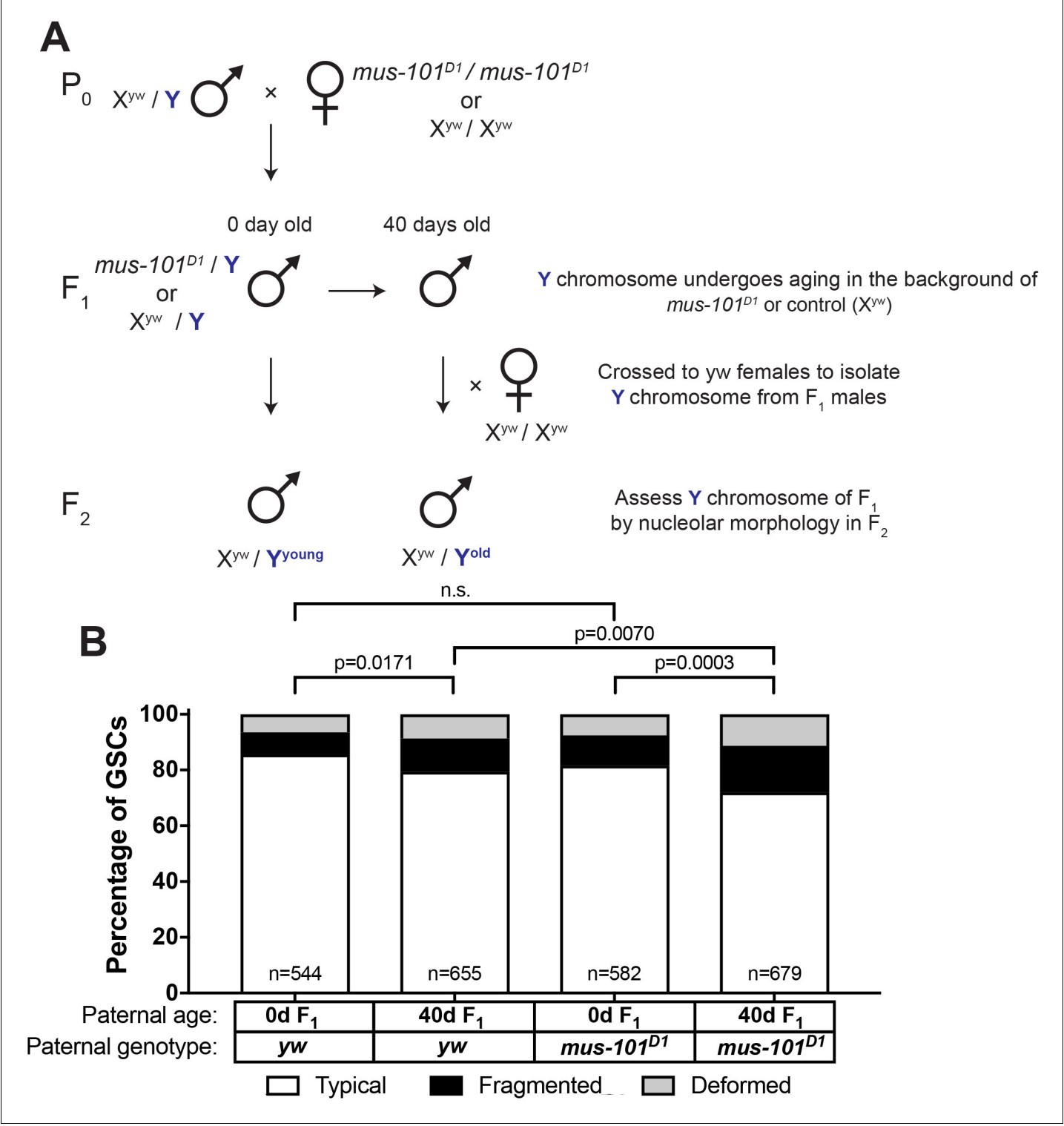

**Figure 7.** *mus-101* is required for rDNA maintenance during aging. (**A**) Mating scheme to compare rDNA loss during aging between *yw* control and *mus-101D1* mutant flies. Males with the same Y chromosome are mated to either *yw* or *mus-101D1* females. *mus-101D1* and *yw* $F_1$ males are mated to young females with the same X chromosome at 0 and 40 days old. All $F_2$ males have X and Y chromosomes from the same source, independent of paternal age or genotype. (**B**) GSC nucleolar morphology in the sons of 0 and 40-day-old *yw* and *mus-101D1* $F_1$, as a percentage of total GSCs scored (n, number of GSCs scored). P-values from chi-squared test between indicated conditions are shown.

DOI: https://doi.org/10.7554/eLife.32421.013

model is consistent with our finding that uninserted copies are selectively lost, potentially providing mechanistic basis to our observations and suggesting that transcribed rDNA copies are preferentially destabilized.

## Nucleolar dominance in *Drosophila melanogaster*

Nucleolar dominance is a phenomenon originally discovered in interspecific hybrids, wherein entire rDNA loci from one species are predominantly activated and those from the other species are silenced (*Chen et al., 1998*; *Chen and Pikaard, 1997*; *Durica and Krider, 1977*; *Preuss and Pikaard, 2007*). However, it was shown that nucleolar dominance also occurs within the *D. melanogaster* males, where Y chromosome rDNA is predominantly expressed, whereas X rDNA is silent (*Greil and Ahmad, 2012*; *Zhou et al., 2012*). It remains unknown whether the nucleolar dominance that occurs within a species vs. that in interspecific hybrids represent the same phenomena or share similar molecular mechanisms. Our study may shed light onto the significance of nucleolar dominance. This study reveals the preferential loss of rDNA copy number from the transcriptionally active Y rDNA locus, whereas the X remains silent. There are a few potential explanations why this might be advantageous: (1) by limiting the transcription of rDNA (thus potential DNA breaks) to one chromosome (i.e. Y chromosome), cells can reduce the risk of deleterious recombination events between the X and Y chromosomes, (2) by maintaining a stable chromosomal locus (X rDNA) for later use, stem cells might be able to extend their life span, delaying the timing of collapse and thus the overall aging. It is tempting to speculate that cells evolved nucleolar dominance to protect their rDNA loci, which are distributed among multiple chromosomes, from deleterious recombination.

It was reported that female neuroblasts exhibit co-dominance between two X chromosome rDNA loci (*Greil and Ahmad, 2012*). We were not able to assess the state of nucleolar dominance in female GSCs for several reasons: (1) after examining multiple *D. melanogaster* strains, we did not detect sufficient SNPs among different X chromosomes and thus we could not perform SNP *in situ* hybridization in females, (2) female GSCs barely showed atypical nucleolar morphology in young or old ovaries: however this could be attributed to constant pairing of two rDNA loci in female GSCs (*Joyce et al., 2013*), thus we could not rely on nucleolar morphology to infer the state of nucleolar dominance. Nonetheless, activating both rDNA loci on two X chromosomes might not impose as serious a risk as in male GSCs, as recombination between two X chromosomes would not lead to deleterious chromosomal rearrangements. Indeed, our inability to find SNPs among X chromosomes from many strains might reflect homogenization of rDNA sequences among X chromosomes through homologous recombination within the population.

In this study, we have adapted SNP *in situ* hybridization (*Levesque et al., 2013*) to assess nucleolar dominance. Previous studies on nucleolar dominance have had to rely on significant sequence differences such as those found in interspecific hybrids, and/or mitotic chromosome spreads (where active rDNA loci can be detected as secondary constrictions of the chromosome or histone H3.3 incorporation) (*Chandrasekhara et al., 2016*; *Greil and Ahmad, 2012*; *Lawrence and Pikaard, 2004*; *McStay, 2006*; *McStay and Grummt, 2008*). These approaches have limited the study of nucleolar dominance to hybrids or certain cell types. Our approach can open up the study of nucleolar dominance to a significantly broader range of species/cell types.

## rDNA copy number maintenance through generations

We showed that rDNA copy number is heritable, wherein old fathers pass a Y chromosome with reduced rDNA copy number to their sons. However, our study revealed that rDNA copy number can recover in the next generation. Although it may be logically deduced that rDNA copy number should not continuously decrease from generation to generation without eventually resulting in complete collapse, the present study is the first to show that rDNA copy number is indeed actively maintained through generations. Our data provide a few important implications in the mechanism of rDNA copy number maintenance. First, it is of particular interest to note that the visible recovery in rDNA copy number was limited to young adults (~first 10 days of adulthood) (*Figure 5*). These results suggest that the rDNA recovery mechanism operates only under certain conditions. It remains unclear if such conditions are developmentally programmed or reflect the limitation of certain cell biological processes that underlie rDNA copy number recovery.

Our study also indicates that the phenomenon classically regarded as 'rDNA magnification' might be a manifestation of a general 'maintenance' mechanism that operates in the population that experiences normal fluctuations in rDNA copy number. We found that mutants that are known to be defective in rDNA magnification also exhibit signs of accelerated destabilization of rDNA during aging (e.g. atypical nucleolar morphology) and fail to restore rDNA in the subsequent generation. These findings suggest that the same molecular mechanisms might underlie rDNA magnification and maintenance.

It has been shown that rDNA copy number changes in response to nutrient conditions, and such copy number changes are inherited to the next generations (*Aldrich and Maggert, 2015*). It has been unknown how this inheritance is achieved. Our results on rDNA copy number changes in germline potentially provide explanation on how rDNA copy number changes are transmitted to the next generation. It awaits future investigation on how nutrient sensing operates in the germline to influence the rDNA copy number to be transmitted to the next generation.

Our findings reveal that tandem rDNA repeats are unstable in *Drosophila* male GSCs, similar to their well characterized instability in yeast, suggesting rDNA loss may occur in other metazoan stem cell populations. Although this instability in germ cells can cause the inheritance of reduced rDNA copies, the germline of young animals has the capacity to restore the lost rDNA copies. These findings suggest that the dynamic contraction and expansion of rDNA loci across generations normally maintains sufficient rDNA copies throughout a population.

# Materials and methods

## Key resources table

| Reagent type (species) or resource | Designation | Source or reference | Identifiers |
| --- | --- | --- | --- |
| Strain, strain background (D. melanogaster) | yw | Bloomington Stock Center | ID_BSC: 1495 |
| strain, strain background (D. melanogaster) | Nopp140-GFP | PMCID: 16158326 | |
| strain, strain background (D. melanogaster) | C(1)RM/C(1;Y)6, y1 w1 f1/0 | Bloomington Stock Center | ID_BSC: 9640 |
| strain, strain background (D. melanogaster) | FM6/C(1)DX, y* f1 | Bloomington Stock Center | ID_BSC: 784 |
| strain, strain background (D. melanogaster) | Df(YS)bb/w1sn1bb*/C(1)RM, y1v1f1 | Bloomington Stock Center | ID_BSC: 4491 |
| strain, strain background (D. melanogaster) | y[1] eq[1]/Df(YS)bb[-] | Kyoto Stock Center | DGRC#: 101–260 |
| strain, strain background (D. melanogaster) | w[1] mus-101[D1] | Bloomington Stock Center | ID_BSC: 2310 |
| strain, strain background (D. melanogaster) | w[1] mei-41[D12] | Bloomington Stock Center | ID_BSC: 6789 |
| strain, strain background (D. melanogaster) | w[1] mus-102[D1] | Bloomington Stock Center | ID_BSC: 2317 |
| antibody | anti-Fibrillarin | Abcam | ID_abcam: ab5821 |
| antibody | anti-Fibrillarin [38F3] | Abcam | ID_abcam: ab4566 |
| antibody | anti-H3K9 dimethyl | Abcam | ID_abcam: ab32521 |
| antibody | anti-vasa | Santa Cruz Biotechnology | ID_SCB: d-26 |
| antibody | anti-Adducin-like 1B1 | Developmental Studies Hybridoma Bank | |
| antibody | anti-vasa | Developmental Studies Hybridoma Bank | |
| antibody | anti-Fasciclin III | Developmental Studies Hybridoma Bank | |

## Fly husbandry and strains

All fly stocks were raised on standard Bloomington medium at 25°C. Unless otherwise stated, flies used for wild-type experiments were the standard lab wild-type strain *yw* (*y$^1$ w$^1$*). C(1)RM/C(1;Y)6, *y$^1$ w$^1$ f$^1$*/0 (Bloomington Stock Center), FM6/C(1)DX, *y* f$^1$* (Bloomington Stock Center)(*Novitski, 1954*), Nopp140-GFP (*McCain et al., 2006*) (a gift of Pat DiMario, Louisiana State University), *Df(YS)bb/ w$^1$sn$^1$bb$^*$/C(1)RM, y$^1$v$^1$f$^1$* (Bloomington Stock Center), *y$^1$eq$^1$/ Df(YS)bb$^-$* (Kyoto Stock Center). Note that we used two Y rDNA deletion chromosomes from different sources: we found *Df(YS)bb$^-$* chromosome from Kyoto stock center has no detectable rDNA on the Y chromosome, whereas the one from Bloomington stock center has reduced rDNA copy number.

## Immunofluorescence staining and microscopy

Immunofluorescence staining of testes was performed as described previously (*Cheng et al., 2008*). Briefly, testes were dissected in PBS, transferred to 4% formaldehyde in PBS and fixed for 30 min. Testes were then washed in PBS-T (PBS containing 0.1% Triton-X) for at least 60 min, followed by incubation with primary antibody in 3% bovine serum albumin (BSA) in PBS-T at 4°C overnight. Samples were washed for 60 min (three 20 min washes) in PBS-T, incubated with secondary antibody in 3% BSA in PBS-T at 4°C overnight, washed as above, and mounted in VECTASHIELD with DAPI (Vector Labs). The following primary antibodies were used: rat anti-vasa (1:20; DSHB; developed by A. Spradling), rabbit anti-vasa (1:200; d-26; Santa Cruz Biotechnology), mouse anti-Fasciclin III (1:200; DSHB; developed by C. Goodman), rabbit anti-Fibrillarin (1:200; Abcam ab5821), mouse anti-Fibrillarin (1:200; Abcam [38F3] ab4566). Images were taken using a Leica TCS SP8 confocal microscope with 63x oil-immersion objectives (NA = 1.4) and processed using Adobe Photoshop software.

## DNA fluorescence *in situ* hybridization

Testes were prepared as described above, and optional immunofluorescence staining protocol was carried out first. Subsequently, fixed samples were incubated with 2 mg/ml RNase A solution at 37°C for 10 min, then washed with PBS-T +1 mM EDTA. Samples were washed in 2xSSC-T (2xSSC containing 0.1% Tween-20) containing increasing formamide concentrations (20%, 40%, then 50% formamide) for 15 min each. Hybridization buffer (50% formamide, 10% dextran sulfate, 2x SSC, 1 mM EDTA, 1 µM probe) was added to washed samples. Samples were denatured at 91°C for 2 min, then incubated overnight at 37°C. Probes used included Cy5-(AATAAAC)$_6$ for detection of the Y chromosome and Cy-3-CCACATTTTGCAAATTTTGATGACCCCCCTCCTTACAAAAAATGCG (a part of 359 bp repeats) for detection of the X chromosome.

In scoring, association of FISH signals (either 359 bp repeat next to the X rDNA or (AATAAAC)$_n$ repeat next to the Y rDNA) with the nucleolus, FISH signal was typically found in the direct proximity to the nucleolus or within the distance smaller than the diameter of FISH signal itself, which was typically less than 0.5 µm. The 'non-associated' FISH signal was far away from the nucleolus. Thus, the distance between associated FISH signal and the nucleolus was always clearly smaller than that between non-associated FISH signal and the nucleolus.

## Determination of X and Y chromosome SNPs

The X chromosome was isolated by crossing experimental XY males with C(1)RM females, generating X/O males lacking the Y chromosome (and the Y rDNA). The Y chromosome rDNA was isolated by crossing experimental XY males with C(1)DX/Y females, which generated C(1)DX/Y females containing our experimental Y and no rDNA on the compound X chromosome. 45S rRNA genes were sequenced using the following primers to identify single nucleotide variants between the two consensus sequences. ITS region: 5'-CTTGCGTGTTACGGTTGTTTC-3' (forward) and 5'- ACAGCATGGACTGCGATATG-3' (reverse). 18S region: 5'-GAAACGGCTACCACATCTAAGG-3' (forward) and 5'- GGACCTCTCGGTCTAGGAAATA-3' (reverse). 28S region: 5'- AGCCCGATGAACCTGAATATC-3' (forward) and 5'- CATGCTCTTCTAGCCCATCTAC-3' (reverse). Sequence alignment was done using ClustalW2.

## SNP RNA *in situ* hybridization

For SNP RNA *in situ* hybridization, all solutions used were RNase-free. Testes were collected in PBS and fixed in 4% formaldehyde in PBS for 30 min. Then testes were washed briefly in PBS, and permeabilized in 70% ethanol overnight at 4°C. Following overnight permeabilization, testes were briefly rinsed in 2xSSC with 10% formamide. Hybridization buffer (prepared according to protocol by LGB Biosearch for Stellaris probes) was prepared with probe (50 nM final concentration) and incubated overnight at 37°C. Following hybridization, samples were washed twice in 2x SSC with 10% formamide for 30 min each and mounted in VECTASHIELD with DAPI (Vector Labs).

Final concentration of each SNP probe was 100 nM, and each mask oligo was 300 nM. Sequences of SNP probes and oligos are provided in *Supplementary file 1*.

## qPCR

Quantitative PCR was carried out using cycling conditions previously described (*Aldrich and Maggert, 2014*) and *Power* SYBR Green reagent (Applied Biosystems). All numbers were normalized to tRNA-K-CTT, a multicopy tRNA gene known to be interspersed throughout the genome, and GAPDH. Primers used are listed in *Supplementary file 2*:

## Mitotic chromosome spreads and fluorescence quantification

Testes were squashed according to previously described methods (*Larracuente and Ferree, 2015*). Briefly, testes were dissected into 0.5% sodium citrate for 5–10 min and fixed in 45% acetic acid/2.2% formaldehyde for 4–5 min. Fixed tissues were firmly squashed with a cover slip then slides were submerged in liquid nitrogen. Following liquid nitrogen, slides were dehydrated in 100% ethanol for at least 5 min. Slides were then treated with 0.1 µg/ml RNase A for 1 hr at room temperature, then dehydrated in 100% ethanol again. Hybridization mix (50% formamide, 2x SSC, 10% dextran sulfate) with 100 ng each probe was applied directly to the slide and allowed to hybridize overnight at room temperature. Then slides were washed 3x for 15 min in 0.2x SSC, and mounted with VECTASHIELD with DAPI (Vector Labs). Sequences for probes used are listed in *Supplementary file 3*.

Images were taken using Leica SP8 confocal microscope, using the setting to detect saturation to ensure that acquired signals were not saturated. Fluorescence quantification was done on merged z-stacks using ImageJ using the Maximum Entropy plugin for automatic thresholding based on the histogram to automatically determine real signal from noise. Using this method, fluorescent probe signal was measured as Integrated Density and compared between the X and Y chromosomes.

## Statistical analysis

For comparison of nucleolar morphologies, significance was determined by chi-squared test using a $2 \times 3$ contingency table (Typical; Deformed; Fragmented). For X rDNA activation by SNP-FISH, because X-only transcription was virtually never detected we simplified the comparison to Y-only rRNA vs both X and Y-rRNA and performed Student's t-tests.

## Acknowledgements

We thank Pat DiMario, Bloomington *Drosophila* Stock Center, Kyoto *Drosophila* Stock Center and Developmental Studies Hybridoma Bank for reagents. We thank the Yamashita lab members, Sue Hammoud and Lei Lei for discussion and comments on the manuscript, and Craig Pikaard for consultation. This research was supported by Howard Hughes Medical Institute. Kevin Lu is supported by National Institute of Aging (F30 AG050398-01A1). Natalie Warsinger-Pepe is supported by NIH Career Training in Reproductive Biology (5T32HD079342-04).

## Additional information

### Competing interests

Yukiko M Yamashita: Reviewing editor, *eLife*. The other authors declare that no competing interests exist.

### Funding

| Funder | Author |
| --- | --- |
| Howard Hughes Medical Institute | Yukiko M Yamashita |
| National Institute of General Medical Sciences | Yukiko M Yamashita |
| National Institute on Aging | Kevin L Lu |
| National Institute of Child Health and Human Development | Natalie Warsinger-Pepe |

The funders had no role in study design, data collection and interpretation, or the decision to submit the work for publication.

## Author contributions

Kevin L Lu, Conceptualization, Formal analysis, Funding acquisition, Investigation, Methodology, Writing—original draft, Writing—review and editing; Jonathan O Nelson, Formal analysis, Validation, Investigation, Writing—original draft, Writing—review and editing; George J Watase, Investigation, Contributed the data (Figure 3—figure supplement 1) and edited the manuscript; Natalie Warsinger-Pepe, Formal analysis, Validation, Writing—review and editing; Yukiko M Yamashita, Conceptualization, Supervision, Funding acquisition, Writing—original draft, Project administration, Writing—review and editing

## Author ORCIDs

Kevin L Lu (iD) https://orcid.org/0000-0003-2677-9537
Jonathan O Nelson (iD) http://orcid.org/0000-0001-9831-745X
George J Watase (iD) http://orcid.org/0000-0001-8250-9027
Natalie Warsinger-Pepe (iD) http://orcid.org/0000-0002-9375-8990
Yukiko M Yamashita (iD) http://orcid.org/0000-0001-5541-0216

## Decision letter and Author response

Decision letter https://doi.org/10.7554/eLife.32421.019
Author response https://doi.org/10.7554/eLife.32421.020

## Additional files

### Supplementary files

• Supplementary file 1. Probe sequences for rRNA SNP *in situ* hybridization.
DOI: https://doi.org/10.7554/eLife.32421.014

• Supplementary file 2. Primer sequences for genomic rDNA qPCR.
DOI: https://doi.org/10.7554/eLife.32421.015

• Supplementary file 3. Probe sequences for rDNA DNA FISH.
DOI: https://doi.org/10.7554/eLife.32421.016

• Transparent reporting form
DOI: https://doi.org/10.7554/eLife.32421.017

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
