## [Decision Letter]

Thank you for submitting your article "Transgenerational dynamics of rDNA copy number in *Drosophila* male germline stem cells" for consideration by *eLife*. Your article has been reviewed by three peer reviewers, one of whom is a member of our Board of Reviewing Editors and the evaluation has been overseen by K VijayRaghavan as the Senior Editor. The following individuals involved in review of your submission have agreed to reveal their identity: Kami Ahmad (Reviewer #2).

The reviewers have discussed the reviews with one another and the Reviewing Editor has drafted this decision to help you prepare a revised submission.

Summary:

Loss of rDNA in aging yeast cells and the presence of damaged rRNA gene copies in many genomes testifies to the challenges of keeping highly active repeated rDNA genes in good working order. *Drosophila* rDNAs are a complex mix of active and inserted genes located on both X or Y chromosomes whose functional gene number varies between strains, with severe deficiencies causing a "bobbed" phenotype. rDNA homogeneity and copy number are maintained at least in part by unequal crossing over. Under certain conditions, rDNA-deficient male germ cells can restore copy number by a "magnification" process whose mechanism and regulation remain poorly understood. In the manuscript, the authors analyze rDNA expression in male germ cells using imaging, FISH, and qPCR to document Y chromosome nucleolar dominance that weakens with age in association with reductions in rDNA copy number. Progeny of old males inherit reduced Y chromosome rDNA copy number but, restore it to normal levels based on cytological assays during the first 10 days of development by an unknown process the authors argue may be magnification.

The work is interesting and if fully validated would represent a significant advance, but at present, the authors' claims are inadequately documented.

Essential revisions:

1) A major problem is that the changes in Y chromosome rDNA copy number are not simply and directly documented. The authors quantify rDNA by Q-PCR in genomic DNA purified from whole testes. Testes contain germ cells, meiocytes, GSCs, gonadal mesoderm (including polyploid cells), etc. This is a very insensitive way to quantify the amount of Y-linked rDNA in GSCs, given the other cell types and the X-linked array signals. Although they attempt a second method to confirm the results, the authors present no data that their allele-specific FISH are quantitative. To critically test the central issue or rDNA copy number changes, the authors should cross young X/Y males with a full complement of rDNA as well as increasingly older males that seem to be losing Y-linked rDNA genes, to rDNA deficient C(1)DX/Y,B females or Df(1)sc4sc8/FM7 females and measure the rDNA levels in young C(1)DX/Y or Df(1)sc4sc8/Y offspring containing only Y-linked rDNA genes. They should observe if these animals begin to show a bobbed phenotype and progressively less rDNA as the age of the X/Y father progressively increases. The same test of Y chromosome rDNA copy number should be done using F_1_ male offspring of an old father, of various ages. In this case, the exclusively Y chromosome rDNA should be deficient when the F_1_ male was young but should return to normal levels during the 1st 10 days of F_1_ adult life.

2) With respect to the FISH assays designed to identify X from Y-linked nucleoli, the criteria used to determine "association" must be described (on and around the second paragraph of subsection “Perturbed nucleolar morphology is associated with transcriptional activation of the normally silent X chromosome rDNA locus.”), especially because the few images presented in Figure 2 show a variance in locations of FISH, fibrillarin staining, etc. This is a concern because the active rDNA are thought to loop into the FC of the nucleolus, while the interspersed inactive cistrons are at the FC/DFC border, and linked chromosome material (including the heterochromatin) may not be found in the nucleolus. Probes to neighboring DNAs (e.g., the 359 satellite) may be some distance away from active cistrons, so proximity is not an adequate indicator of transcriptional activity.

3) The relational of this work to magnification is currently exaggerated and needs to be accurately discussed. First, a summary of previous work on rDNA magnification including recent work should be given in the introduction. Second, the authors must recognize that there are no known "rDNA magnification" genes exclusively involved in this process. Hence, conclusions such as: "This result indicates that rDNA magnification contributes to germline rDNA maintenance during normal aging" are unjustified. Third, the authors should specifically include a discussion reconciling the differences in frequencies of magnification observed in previous studies (>1%) from those reported here (100%). Fourth, the authors should include a discussion of the previous observations that only some rDNA array-containing chromosomes are subject to magnification (and only in some genetic conditions), and how that would bear on their model.

4) It seems to be a misnomer to describe the changes in nucleolar appearance as aberrant nucleolar morphology. The FISH mapping shows that these changes are not a nucleolar defect but are simply the activation of two loci.

5) The authors do not address the very interesting question of whether the nucleolar phenomena and rDNA copy number changes they observed are confined to germ cells. Is there nucleolar dominance, loss of genes with aging, recovery in F_1_ adults in somatic hub cells and cyst stem cells, like in GSCs? In particular, does the reduced level of Y rDNA inherited from aged fathers persist in the soma of F_1_ animals, even though rDNA copy number is restored in the germline?

6) Figure 5: Can't see the X and Y signals in panel B, F1 10d.

7) Subsection “Destabilization of rDNA loci during aging in *Drosophila* male GSCs.” Why would activation of additional rDNA genes to compensate for lost genes, leaving the total number of active genes intact, "increase the chance of replication transcription collisions and generate a vicious cycle? The argument is not clear here.

8) The structure of the Y and X chromosome rDNAs is not fully described and affect some of the inferences made in this manuscript. The authors describe loss of 45S cistron copies with age, but no loss of R1/R2 retroposons. Are there R1/R2 insertions on the Y chromosome (implying preferential loss), or are all the copies on the X chromosome?

9) The data in Figure 3 and the fourth paragraph of subsection “Y chromosome rDNA copy number decreases in the male germline during aging.” details the loss of rDNA cistrons by qPCR. It would be worth converting these numbers into assignments for each chromosome, i.e. in a wildtype male 34% of 18S sequence is on the X and 66% on the Y. Then, after 40 days, there is (if the X doesn't change) 34% on the X and 18% remaining on the Y (loss of two thirds of the locus).

9) Multiple graphs are interrupted to emphasize the small changes in nucleolar morphology frequencies. This may overstate the size of the effect. There seems to be some discordance of the measurements of rDNA loss by qPCR and the more subtle changes in nucleolar appearance. Is this possibly due to a threshold at which nucleolar dominance is lost?

10) The age effect described predicts that there would be loss or rDNA cistrons from the X chromosome when combined with a Df(YS)bb; is this the case? Overall, more details on the structure of the "partially deleted" Df(YS)bb would be useful.

11) The authors under-explain the results by Ahmad and Hartl and Eickbush, whose groups all saw that not all chromosomes or genetic background manifest nucleolar dominance. Further, they do not mention that in plants nucleolar dominance is a transient phenomenon, often lost after the first week of development. These findings may bear on the authors' interpretation and certainly bear on the "null hypotheses" that loss of Y-dominance is due to Y-linked rDNA loss.

12) Eickbush has done structural studies of the R2 distribution within the arrays and finds them mostly distributed throughout. How can all the copies of active rDNA be lost without reducing the copies of the interspersed R2 inserted copies?

[Editors' note: further revisions were requested prior to acceptance, as described below.]

Thank you for resubmitting your work entitled "Transgenerational dynamics of rDNA copy number in *Drosophila* male germline stem cells" for further consideration at *eLife*. Your revised article has been favorably evaluated by K VijayRaghavan (Senior editor), a Reviewing editor, and three reviewers.

The manuscript has been significantly improved and no further experiments are needed. However, some important changes in the presentation need to be addressed before acceptance and publication, as outlined below:

The revised version of the paper "Transgenerational dynamics of rDNA copy number in *Drosophila* male germline stem cells" has addressed most of the issues raised previously in the opinion of a majority of the reviewers. In particular, the assays clearly demonstrate that rDNA copy number and nucleolar usage changes in a regular way within male germ cells over the course of a lifecycle. This discovery is significant because rDNA copy number is currently thought to be much more stable, and all rDNA genes, if not inserted, are currently thought to be equivalently regulated. The work reported here is inconsistent with one or both of these assumptions and will stimulate new interest and advancements in this area.

However, the authors do not demonstrate how the observed germline changes relate to somatic rDNA levels and phenotype. They assume that the changes they observe in germ cell rDNA copy number also affect somatic cell rDNA copy number and could in an extreme case generate a defective NO. Since somatic cells could compensate significantly for the observed low level rDNA copy number variation by differential replication, and since no mutant chromosome derivatives or bobbed flies have been recovered, this is unjustified. Consequently, the final version of the paper should focus on the observed changes in germline rDNA copy number and nucleolar usage. How cyclic age dependent losses and recovery affect the soma and directional rDNA evolution, if it does so at all, remains unclear. As a result, it is premature to speculate so extensively on how these small germline rDNA copy number changes might relate to rDNA magnification. Such speculation should be cut back to 1 or two sentences and saved for a publication in which the copy number and phenotypic expression of rDNA in somatic cells is experimentally addressed.

In addition, the line-breaks in the graphs should be removed as the authors agreed in their response.

Second, the authors should acknowledge that the FISH experiments have not been shown to be fully quantitative.

*Reviewer #1:*

In the revised version and the response to reviewers, the authors have cogently addressed my concerns regarding some weaknesses and reviewer misunderstandings in the first version of this paper. This is a highly original work that addresses a central but neglected topic, namely the role of rDNA instability in stem cells during aging. The authors developed cytologically based methods that allow X and Y nucleolar activity to be visualized independently in stem cell. The in-situ hybridization data provide convincing corroboration of the PCR data that rDNA gene copy number is changing during aging and undergoing restoration in you F_1_ males. This paper will stimulate research in this field to answer many remaining questions concerning the developmental timing and mechanism of rDNA restoration, and on the epigenetic mechanisms that program rRNA gene activation and repression.

*Reviewer #2:*

The revised manuscript "Transgenerational dynamics of rDNA copy number in *Drosophila* male germline stem cells" extends analysis of changes in rDNA expression using SNPs in transcripts, and includes new data using FISH to determine gene copy number on deletion Y chromosomes. These data do help document changes in gene copy number but there are some details that should be included. The authors use two deletion Y chromosomes to demonstrate that FISH is sensitive to copy number differences. However, copy numbers of these chromosomes (determined by some other method) should be reported. The authors seem to be using two different deletion Y chromosomes, but the names of these chromosomes are confusing (Df(YS)bb is different from Df(YS)bb-?).

A central issue that was raised in the first review was whether aging throws bobbed-deficient Y chromosomes, that should be apparent by some frequency of bobbed phenotypes in progeny. The authors allude that they have some data on this, but this is not clearly described. They should include a more thorough description and data supporting the statement in subsection “rDNA copy number maintenance through generations” that "we barely observed bobbed phenotype even among the sons of very old fathers". Given the extent of copy number loss the authors are estimating, such bobbed progeny are expected.

Reviewer #3:

This is a re-review of "Transgenerational dynamics of rDNA copy number in *Drosophila* male germline stem cells," *eLife*, submitted by Yukiko Yamashita and colleagues.

My concerns remain, even after the authors sought to clarify their findings.

First, I remain very concerned that the vast literature on rDNA magnification remains unaddressed, even though those data are at odds with the observations made here. That concern is more profound because the measurements of rDNA loss here are not direct and simple: the authors use nucleolar shape, Y:X ratio, unvalidated FISH, and qPCR quantification on one tissue relatively late in development (eclosed adult testes) as proxy for rDNA copy number in sperm; more direct and more reliable assays are available.

The authors assert that rDNA counts drop with aging males, that those lower counts are still detectable (and are to the same degree) in testes of newly-eclosed males, they recover soon, and are lost again (presumably due to aging). The essential experiment that was called for in the first review was to measure loss in the whole soma of the progeny (in some type of rDNA[0]/Y flies – males of sc[4]sc[8]Y or females of DX,rDNA[0]/Y). The authors state that they did not observe any bobbed phenotype, which is perhaps expected if the original Y had a large number of rDNA copies, however losses should be easily detectable in whole animals using qPCR (as Aldrich and Maggert did), or the authors could start with shorter rDNA array containing Ys, such as one of the bobbed alleles to which they have access, and measure loss based on bobbed phenotype. The justification for asking this is simple: the Ys from old males should have reduced rDNA copy number, obvious in all of the soma of the offspring. In the case of the bobbed literature, this would have been noticed (by > dozens of reports) as an increase in the bobbed expressivity or penetrance, which was never reported; in fact rDNA copy number is remarkably robust fly-to-fly and generation-to-generation, which is at odds with the observations here. Either those previous studies are wrong, or they did not do the crosses in a way to detect such a remarkable variation in rDNA copy number, or the observations here are not detecting rDNA loss. Whichever the answer is, this has to be resolved prior to accepting the author's conclusion of their data (Points 1, 3, 10).

As an aside, in terms of rDNA copy number changes, nothing in the authors' data rule out that they are observing a process of endoreduplication/polytenization in a subset of cells in the testes (as they allude to in their response to Point 5). If that is the case, the apparent "losses" and "gains" would not bear on the sperm haploid genomes, which would be consistent with both the authors' measured effects *and* the known genetic stability from others' work on bobbed magnification. Without looking in the soma of the offspring, they cannot know. This experiment, in my mind, remains essential.

Second, I am unconvinced by the authors' assurance that their FISH is quantitative. Their care to not saturate the signal is fine, but they do not show any data that it is quantitative (which would require determining a lowest-threshold, a linearity of response between that threshold and saturation, and ample controls/experiments showing that photobleaching, preferential binding, etc are not issues). I think that a lower threshold must exist since the authors cannot detect their hypothetical extrachromosomal rDNA (Lines 371-374). FISH is not intrinsically quantitative and cannot be used as such unless great pains are taken. Hence, treating or calling it quantitative is not appropriate (Point 1).

Related to this, I'd still like to know how exclusive *and* abundant these SNPs are to the two (X-linked and Y-linked) rDNA arrays (Point 8).

Third, I still do not understand how the overall structure of the Y-linked array is consistent with the authors' model. If R1- and R2- inserted rDNA copies are interspersed with uninserted rDNA, I cannot imagine a damage-and-repair mechanism (that involves intrachromosomal HR-based recombination) that does not remove intervening rDNA. Either way, the structure of the Ys (i.e., degree of interspersion/clustering of R1- and R2- elements in the entire rDNA gene cluster) being used in this study should be included, as we asked (Points 8 and 12).

In general, I am not satisfied with the discussion of magnification and how that bears on these studies. I am not convinced they are the same thing, and I do not understand how the authors envision their proposed phenomenon and magnification are related. In subsection “rDNA copy number maintenance through generations”, they propose unequal sister exchange as a mechanism, however the authors should explain the fate of the other (shortened) product of such an exchange. Are they detectable in their data? (Points 3, 11).

I continue to be uncomfortable with the data presentation, with line breaks accentuating the magnitude of effects. Breaks are conventionally used to graph data with vastly different values on the same graph, where here they are used consistently but have the effect of making smaller effects seem larger. Graphs should give a visual representation of the size of effects, but the use in this paper overemphasizes them (sometimes to a very large degree). This change remains essential (Point 9).

The question as to whether an X, in X/Ybb males, is also subject to these losses seems central to the authors' model. It can – and should be – addressed experimentally (Point 10).

These above experiments were originally denoted as "essential," but they were not done in this revision. It is my opinion that this study – if correct – could overturn decades of research on rDNA copy number determination, rDNA stability and changes, and the work on rDNA magnification. It is thus critical that the appropriate experiments be done to rule out alternatives and to pursue the specific predictions of these proposed models.

---

## [Author Response]

Summary:Loss of rDNA in aging yeast cells and the presence of damaged rRNA gene copies in many genomes testifies to the challenges of keeping highly active repeated rDNA genes in good working order. Drosophila rDNAs are a complex mix of active and inserted genes located on both X or Y chromosomes whose functional gene number varies between strains, with severe deficiencies causing a "bobbed" phenotype. rDNA homogeneity and copy number are maintained at least in part by unequal crossing over. Under certain conditions, rDNA-deficient male germ cells can restore copy number by a "magnification" process whose mechanism and regulation remain poorly understood. In the manuscript, the authors analyze rDNA expression in male germ cells using imaging, FISH, and qPCR to document Y chromosome nucleolar dominance that weakens with age in association with reductions in rDNA copy number. Progeny of old males inherit reduced Y chromosome rDNA copy number but restore it to normal levels based on cytological assays during the first 10 days of development by an unknown process the authors argue may be magnification.The work is interesting and if fully validated would represent a significant advance, but at present, the authors' claims are inadequately documented.

Before getting into the details of the response, first we would like to summarize two major points in addressing reviewers’ concerns.

Major point 1. We document that rDNA is reduced during aging, which is the first clear demonstration of rDNA reduction in stem cells during aging in multi-cellular organisms. Reviewers felt that we have not adequately shown that rDNA loss occurs exclusively occur from Y chromosome. We do not claim or believe that rDNA copy number reduction occurs exclusively on the Y chromosome. Our results only show ‘preferential’ loss of Y rDNA, which we speculate (based on our results and established knowledge in the literature) to be caused by transcriptionally active state of Y rDNA. This does not indicate that silent rDNA (on X) never loses the copy number. In addition, X rDNA also becomes active at some point during aging, thus likely start losing the copy number. Importantly, our major conclusion on rDNA copy number reduction in aging does not lose its impact if it’s not exclusively from Y rDNA. We edited throughout the text to clarify this point to ensure that we do not indicate that rDNA loss occurs exclusively on Y rDNA.

Major point 2. We provide evidence that reduced rDNA copy number can recover in the subsequent generation (specifically during young age), which we call a ‘maintenance mechanism’, and we propose that classically-observed ‘magnification’ phenomenon may be the manifestation of this ‘maintenance mechanism.’ Reviewers felt that it is strange that some observations do not match between rDNA magnification from previous studies and rDNA maintenance mechanism in our study. Our proposal is that magnification is an extreme case that utilizes ‘maintenance mechanism’, and that ‘regular’ maintenance mechanism operates within the range without showing extremity of magnification (i.e. extreme rDNA loss to yield bobbed phenotype). We do not intend to claim that the maintenance mechanism we describe here is a synonymous of rDNA magnification: instead, rDNA magnification is manifestation of rDNA maintenance mechanism, which may not be equally efficient when recovering from extreme degrees of rDNA loss. We have clarified this point throughout the text.

Essential revisions:1) A major problem is that the changes in Y chromosome rDNA copy number are not simply and directly documented. The authors quantify rDNA by Q-PCR in genomic DNA purified from whole testes. Testes contain germ cells, meiocytes, GSCs, gonadal mesoderm (including polyploid cells), etc. This is a very insensitive way to quantify the amount of Y-linked rDNA in GSCs, given the other cell types and the X-linked array signals. Although they attempt a second method to confirm the results, the authors present no data that their allele-specific FISH are quantitative. To critically test the central issue or rDNA copy number changes, the authors should cross young X/Y males with a full complement of rDNA as well as increasingly older males that seem to be losing Y-linked rDNA genes, to rDNA deficient C(1)DX/Y,B females or Df(1)sc4sc8/FM7 females and measure the rDNA levels in young C(1)DX/Y or Df(1)sc4sc8/Y offspring containing only Y-linked rDNA genes. They should observe if these animals begin to show a bobbed phenotype and progressively less rDNA as the age of the X/Y father progressively increases. The same test of Y chromosome rDNA copy number should be done using F_1_ male offspring of an old father, of various ages. In this case, the exclusively Y chromosome rDNA should be deficient when the F_1_ male was young but should return to normal levels during the 1st 10 days of F_1_ adult life.

Concerns raised here are important, because these data are the most critical foundation of our proposal in this study. First, as we described in our ‘major point 1’ above, the importance of the present study does not rely on rDNA reduction being exclusively from Y rDNA. qPCR is not intended to show Y-specific loss of rDNA. qPCR has been used successfully by Maggert lab to detect changes in rDNA copy number (Paredes and Maggert, 2009), and we believe that the qPCR results here establishes the overall loss of rDNA in the testis, which is primarily composed of germ cells derived from GSCs.

The weakness due to the use of the whole tissue was complemented by DNA FISH, which surely focuses on GSCs and their progeny. Please note that the DNA FISH method to quantify rDNA on chromosomes were not ‘allele specific’ unlike the RNA FISH in this study, and the probes should hybridize equally to both X and Y rDNA loci (as described in the Materials and methods section and Supplementary file 3, 18S rDNA on mitotic chromosomes were detected by Stellaris probes, comprised of 48 fluorescently-labelled oligonucleotides). We distinguished X vs. Y rDNA loci by adding Y-specific marker (AATAAAC)_n_, which is located right next to the Y rDNA (although we can easily distinguish X and Y chromosomes by their mitotic morphology, we used this (AATAAAC)_n_ signal as a definitive marker to tell apart X vs. Y). We used our microscope setting to detect ‘signal saturation’, and all of images for quantification were obtained under the condition where no signal was saturated (this detail was added to the method section). Thus, pixel intensity analysis comparing X vs. Y rDNA ratio is expected to be very accurate. Combined with the results that we can detect clear reduction in Y rDNA in the partial Y rDNA deletion *Df(YS)bb* mutant, and complete loss in the *Df(YS)bb-* deficiency (now in Figure 3—figure supplement 1), we believe that this method is sufficiently quantitative.

We do not believe that assessing rDNA loss by bobbed phenotype observation is a sensitive method. While this is the most well-established characterization of rDNA insufficiency, changes in rDNA copy number can occur without the bobbed phenotype arising. It is known that the *Drosophila* stocks can have varying amount of rDNA (80-600 copies per genome) (Mohan and Ritossa, 1970) and it was described that bobbed phenotype arises when rDNA copy number is lower than 130 in total (Ritossa et al., 1966), although this number may vary depending on genetic background. This indicates that rDNA copy number can fluctuate without showing bobbed phenotype. As the Figure 3 suggests, the reduction in Y rDNA is around ~50% in aged males, the level that would not necessarily show bb phenotype even when combined with Xbb chromosome (rDNA complete loss). Indeed, we have already conducted this cross (crossing aging Y chromosomes into the background of Xbb chromosomes) and we did not see detectable increase in bobbed phenotype, indicating that rDNA loss during aging is not significant enough to cause bobbed phenotypes.

Instead, to better establish the heritable effects of germline rDNA copy number loss during aging, we crossed young vs. old fathers to young mothers to conduct FISH analysis in the sons, where young mother-derived X chromosome serve as a standard to compare ‘young Y’ vs. ‘old Y’ (as the ratio of Y^young^:X^young^ vs. Y^old^:X^young^, where X^young^ coming from the same source of females. The results confirmed that Y chromosomes inherited from old fathers have reduced rDNA copies compared to Y chromosomes from young fathers. This data is now included in Figure 4. We hope that the added experiments, the clarification of the text and methodology address reviewers’ concerns.

2) With respect to the FISH assays designed to identify X from Y-linked nucleoli, the criteria used to determine "association" must be described (on and around the second paragraph of subsection “Perturbed nucleolar morphology is associated with transcriptional activation of the normally silent X chromosome rDNA locus.”), especially because the few images presented in Figure 2 show a variance in locations of FISH, fibrillarin staining, etc. This is a concern because the active rDNA are thought to loop into the FC of the nucleolus, while the interspersed inactive cistrons are at the FC/DFC border, and linked chromosome material (including the heterochromatin) may not be found in the nucleolus. Probes to neighboring DNAs (e.g., the 359 satellite) may be some distance away from active cistrons, so proximity is not an adequate indicator of transcriptional activity.

We agree that our FISH probes (359 repeat or AATAAAC repeat) do not actually represent the localization of rDNA itself. We also agree that distance between the FISH signal and the nearby nucleolus can vary to some extent as the reviewers pointed out. However, we did not have any difficulty in assigning which FISH signal (359 or AATAAAC) is closer to the nucleolus for the following reasons: most of the case, each nucleolus was juxtaposed to a FISH signal with no gap in between at all, or with the gap no larger than the FISH signal’s diameter itself (typically ~0.5µm), whereas the distance between the nucleolus and the other (non-associated) FISH signal was more than a few microns (this detail was added to the method). We did not encounter any ‘close calls’, where a nucleolus is equally far from X and Y FISH signals. If the reviewers’ concern is indeed the case, we would have observed cases where a nucleolus does not have any nearby FISH signals. Instead, we observed that every single nucleolus was found to be associated with a FISH signal, making us confident that rDNA and the juxtaposing heterochromatin locus are not separated far away. Therefore, we believe that it is highly unlikely that we are assigning a nucleolus to a wrong rDNA locus.

3) The relational of this work to magnification is currently exaggerated and needs to be accurately discussed. First, a summary of previous work on rDNA magnification including recent work should be given in the introduction. Second, the authors must recognize that there are no known "rDNA magnification" genes exclusively involved in this process. Hence, conclusions such as: "This result indicates that rDNA magnification contributes to germline rDNA maintenance during normal aging" are unjustified. Third, the authors should specifically include a discussion reconciling the differences in frequencies of magnification observed in previous studies (>1%) from those reported here (100%). Fourth, the authors should include a discussion of the previous observations that only some rDNA array-containing chromosomes are subject to magnification (and only in some genetic conditions), and how that would bear on their model.

We appreciate these constructive comments that helped improve the clarity of the manuscript. We have rewritten the section describing rDNA magnification and the involvement of *mus^-1^01* and *mei-41* genes in rDNA magnification to better clarify the points mentioned here by the reviewers. The manuscript now emphasizes that these genes are not necessarily ‘rDNA magnification’ genes, but instead these genes, which also have functions in other cellular processes, have been shown to be required for rDNA magnification. We indicate that the conceptual similarities between rDNA magnification and the recovery of rDNA in F1s and the requirement of the same genes for both phenomena suggests that the same mechanisms may underlie both processes and thus germline rDNA maintenance in general. Our introduction to rDNA magnification now includes that the majority of magnification is primarily observed in the offspring of males with large deletions of Y rDNA. This is an important feature of our conclusion that the same mechanisms underlie both phenomena, as they both occur in the germline of males with reduced Y rDNA. The revisions to the text can be found in subsection “Recovery of GSC nucleolar morphology depends on the homologous recombination repair pathway”.

Regarding the third point, we realize that the text in the previous version of the manuscript was not clear enough and confused the reviewers. Our hypothesis (based on the data presented in this study) is that rDNA copy number is maintained within a ‘normal’ range (without undergoing severe reduction to the extent of showing bobbed phenotype) through generations by the ‘maintenance’ mechanism, as discussed above (our ‘major point 2’ at the beginning of the response letter). In contrast, magnification, which is defined by the recovery from the severe reduction (showing bb phenotype) to a phenotypically-normal range, is the process that likely pushes the limit of ‘maintenance’ mechanism. Therefore, it is plausible that the magnification does not happen as efficiently as regular ‘maintenance’ recovery. We have clarified this issue in the revised text in subsection “rDNA copy number maintenance through generations”.

4) It seems to be a misnomer to describe the changes in nucleolar appearance as aberrant nucleolar morphology. The FISH mapping shows that these changes are not a nucleolar defect but are simply the activation of two loci.

The term ‘abnormal’ nucleolar morphology is not meant to convey that there is a defect in the nucleolus itself, but simply that this morphology is different from the ‘normal’ morphology (i.e. single, round nucleolus morphology) that is most common. To prevent confusion, we have edited the text to instead describe the deformed and fragmented morphology as ‘atypical.’

5) The authors do not address the very interesting question of whether the nucleolar phenomena and rDNA copy number changes they observed are confined to germ cells. Is there nucleolar dominance, loss of genes with aging, recovery in F_1_ adults in somatic hub cells and cyst stem cells, like in GSCs? In particular, does the reduced level of Y rDNA inherited from aged fathers persist in the soma of F_1_ animals, even though rDNA copy number is restored in the germline?

We also find the possibility of changes in rDNA in somatic stem cells to be very interesting. However, we consider the rDNA of the germline to be the most relevant, since these are the only cells that contribute genetic information for subsequent generations and our report concerns transgenerational inheritance of the rDNA dynamics. Also, we consider that the study of somatic stem cells might not provide relevant information that is worth comparing with the results obtained from germline for a few reasons described below.

In an ongoing project in our laboratory, we found that nucleolar dominance indeed happens in somatic tissues of males during development (Y dominant in gut, fat body, larval brain etc.). However, studying nucleolar dominance during aging in these somatic tissues is not relevant to the present study, because 1) some somatic cells (such as neuroblast) do not persist into adulthood (thus the effect of aging cannot be studied), 2) other somatic cells stop proliferating, and thus any observations made in such cells cannot be interpreted in parallel with GSCs, where our interest is in aging phenotype likely caused by repeated cell division cycles, 3) yet other somatic cells undergo polyploidization, confounding the interpretation of most of experiments that were employed in this study (qPCR, FISH etc.). Because of these confounding factors, and because the knowledge obtained from the somatic cells cannot be utilized to better interpret the results of the present study, we consider this to be beyond the scope of present study.

6) Figure 5: Can't see the X and Y signals in panel B, F1 10d.

We have altered the figure to include arrows indicating the X signal. Also, we apologize that we had mistakenly circled (marked) GSCs whose nucleolus cannot be seen (being out of the focal plane). Now we removed those circles and only the relevant GSCs are marked.

7) Subsection “Destabilization of rDNA loci during aging in Drosophila male GSCs.” Why would activation of additional rDNA genes to compensate for lost genes, leaving the total number of active genes intact, "increase the chance of replication transcription collisions and generate a vicious cycle? The argument is not clear here.

The rationale for this argument is based on the assumption that the same number of rDNA copies would need to be transcriptionally active in all cells of the same type (GSCs in this case), no matter how many rDNA copies there are in each individual cell (e.g. 150 copies to be transcriptionally active out of total 200 copies (75%) vs. 150 copies to be active out of total 450 copies (33%)). In this scenario, cells with fewer rDNA copies have a higher proportion of active rDNA, as the denominator in the ratio is lower than in cells with more rDNA, while the numerator is constant. The likelihood of a collision between transcription and replication machinery is a function of the frequency that any given region is being transcribed or replicated. As the portion of rDNA that is being transcribed increases, the frequency of transcription across all rDNA, on average, increases, meaning the likelihood for collision with replication machinery also increases, even though there is no change to the rate of replication. Since collisions between transcription and replication machinery have been proposed to induce DNA damage that can cause further rDNA copy loss, this creates the potential for a ‘vicious cycle.’ We propose there is the potential that rDNA loss can become accelerated because as copies are lost, the portion of rDNA that is transcriptionally active increases, increasing the probability of transcription-replication collisions, which cause further rDNA loss, thus causing a greater increase in the portion of active rDNA, and creating more opportunity for more rDNA loss. We have edited the text at in subsection “Destabilization of rDNA loci during aging in *Drosophila* male GSCs” to better clarify this argument within the discussion.

8) The structure of the Y and X chromosome rDNAs is not fully described and affect some of the inferences made in this manuscript. The authors describe loss of 45S cistron copies with age, but no loss of R1/R2 retroposons. Are there R1/R2 insertions on the Y chromosome (implying preferential loss), or are all the copies on the X chromosome?

We have done qPCR from animals, which harbors Y rDNA only or X rDNA only to determine the relative copy number of R1 and R2 on X vs. Y chromosomes. Interestingly, we found that R2 copy number is similar between the X and Y chromosome, whereas R1 is much more abundant on the X chromosome than the Y (see Author response image 1). This finding indicates that the Y chromosome has many more uninserted rDNA copies than the X chromosome. However, this data is uninformative to our hypothesis that transcriptionally active rDNA copies are preferentially lost, since it is unclear if the more abundant uninserted copies on the Y may or may not influence Y dominance, and any assumption that it does influence Y dominance would be purely speculative. Since this data neither supports nor weakens our hypothesis, we consider it to be unessential to main analysis in this report and distracting from those points, and decided it is better to leave this data unpublished until the context of this result can be better understood in the future. If, however, the editor/reviewers do think that including this data would be critical for publication, we would be happy to include it with this manuscript.

9) The data in Figure 3 and the fourth paragraph of subsection “Y chromosome rDNA copy number decreases in the male germline during aging.” details the loss of rDNA cistrons by qPCR. It would be worth converting these numbers into assignments for each chromosome, i.e. in a wildtype male 34% of 18S sequence is on the X and 66% on the Y. Then, after 40 days, there is (if the X doesn't change) 34% on the X and 18% remaining on the Y (loss of two thirds of the locus).

Although it is theoretically possible to combine qPCR data and the FISH data (to assess the XY ratio) to convert to numbers such as ‘33% on X, 66% on Y’, we do not think that this conversion would provide an accurate estimation for multiple reasons. First, our data do not conclude that copy number loss is ‘exclusively’ from Y rDNA (our ‘major point 1’ described at the beginning of the response letter). We think that our data only suggest that the copy number loss is ‘preferentially’ from Y (due to its transcriptional activity). Note that transcriptional activity only accelerates instability but is not the sole cause of instability. Thus, silent X rDNA is expected to lose its copy number albeit at a lower late. During aging, as nucleolar dominance is compromised, X rDNA becomes active, and thus likely accelerates the rate of instability: this means that X rDNA’s rate of instability is not constant during the course of GSCs’ aging. Because X rDNA copy number cannot be assumed to be constant during aging for these reasons, simply combining qPCR data and FISH data would not provide an accurate estimation of the total amount of rDNA on each chromosome. We have changed the text in subsection “Y chromosome rDNA copy number decreases in the male germline during aging” to clarify this distinction.

9) Multiple graphs are interrupted to emphasize the small changes in nucleolar morphology frequencies. This may overstate the size of the effect. There seems to be some discordance of the measurements of rDNA loss by qPCR and the more subtle changes in nucleolar appearance. Is this possibly due to a threshold at which nucleolar dominance is lost?

Please note that we consistently made the break in ALL graphs describing percentage of nucleolar morphology or nucleolar dominance (X and Y expression), for ease of reading and greater transparency of our data (by making thin stacks or error bars more visible). Exact p-values are provided to all relevant data sets, and the number is clearly indicated on Y axis. Thus, we believe that these are fair and even representation of the data under all conditions. We did not do this to mislead or exaggerate the effect of our findings (if we have done this manipulation to only a subset of graphs, it would have been misleading, but it is not what we did). If it is critical to represent the data without the breaks, we can remove them, but we believe that the breaks make our data more digestible for our readers.

10) The age effect described predicts that there would be loss or rDNA cistrons from the X chromosome when combined with a Df(YS)bb; is this the case? Overall, more details on the structure of the "partially deleted" Df(YS)bb would be useful.

Indeed, X/Df(YS)bb flies would have active X rDNA, which is clearly indicated in Figure 3. And we certainly predict that X rDNA would be more destabilized. However, we would like to clarify that we did not/do not claim that silent X rDNA is entirely stable. The core message of the present study is dynamic nature of rDNA copy number (decrease during aging, recovery in the next generation), and we do not think this is X or Y chromosome-specific phenomenon. Transcriptional state accelerates but not solely determines the instability - thus, Y rDNA is more affected than X rDNA, but not exclusively. Indeed, the data in Figure 4 suggest that X chromosome is also somewhat compromised during aging, and we had discussed that X rDNA is also likely affected during aging. We have now clarified this point in the revised text not to leave the impression that rDNA instability occurs exclusively on the Y chromosome.

Regarding the nature of the Df(YS)bb chromosome, it is the Y rDNA deficiency stock listed at the Bloomington *Drosophila* Stock Center, and is previously characterized having insufficient Y rDNA for viability when X rDNA is disrupted (Cline, 2001). This reference is now included in the text. The nature of ‘partial deletion’ of this chromosome is described in this study by DNA FISH, which shows clear reduction (but not complete deletion) of Y rDNA (Figure 3).

11) The authors under-explain the results by Ahmad and Hartl and Eickbush, whose groups all saw that not all chromosomes or genetic background manifest nucleolar dominance. Further, they do not mention that in plants nucleolar dominance is a transient phenomenon, often lost after the first week of development. These findings may bear on the authors' interpretation and certainly bear on the "null hypotheses" that loss of Y-dominance is due to Y-linked rDNA loss.

Please note that (Zhou et al., 2012) by Hartl and Eickbush groups provided quite strong evidence that Y rDNA is dominant over X rDNA in all genetic backgrounds examined (although they found that position effect variegation shows variance based on Y rDNA contents). (Greil and Ahmad, 2012) have presented a few Y chromosome variants that exhibit co-dominance, where rDNA copy number cannot account for the observed co-dominance. However, considering the fact that the very same chromosome from a particular stock is undergoing aging in our aging experiments, we consider it to be highly unlikely that ‘background’ changes occur to these chromosomes, influencing the nucleolar dominance.

In regards to developmental changes in nucleolar dominance in plants, it is established that while co-dominance occurs during early development in both *Arabidopsis* hybrids and *A. thaliana* non-hybrid development, a subset of rRNA genes are silenced during early vegetative development (days 10-14) the dominance is then maintained (Earley et al., 2010; Pontes et al., 2007). This suggests that loss of established nucleolar dominance is in fact uncommon unless rDNA is disrupted.

12) Eickbush has done structural studies of the R2 distribution within the arrays and finds them mostly distributed throughout. How can all the copies of active rDNA be lost without reducing the copies of the interspersed R2 inserted copies?

Recent work from Thomas Eickbush’s lab proposes a model that may explain this bias in the rDNA copies lost during aging. (Zhou et al., 2013) suggests that transcription of rDNA preferentially occurs in contiguous blocks of uninserted rDNA copies, and that intra-chromosomal exchanges that cause rDNA loss only occur within these transcribing blocks, thus preferentially causing loss of uninserted rDNA copies. Therefore, preferential loss of uninserted copies (as shown in Figure 3) is consistent with the current knowledge of the field. We have added to the text in subsection “Destabilization of rDNA loci during aging in *Drosophila* male GSCs” to include in our discussion how our data is consistent with this model for rDNA loss

[Editors' note: further revisions were requested prior to acceptance, as described below.]

The manuscript has been significantly improved and no further experiments are needed. However, some important changes in the presentation need to be addressed before acceptance and publication, as outlined below:The revised version of the paper "Transgenerational dynamics of rDNA copy number in Drosophila male germline stem cells" has addressed most of the issues raised previously in the opinion of a majority of the reviewers. In particular, the assays clearly demonstrate that rDNA copy number and nucleolar usage changes in a regular way within male germ cells over the course of a lifecycle. This discovery is significant because rDNA copy number is currently thought to be much more stable, and all rDNA genes, if not inserted, are currently thought to be equivalently regulated. The work reported here is inconsistent with one or both of these assumptions and will stimulate new interest and advancements in this area.

We are grateful that the majority of reviewers agreed on the significance of our work and its suitability for publication.

However, we are afraid that the perceived ‘inconsistency with the literature’ mentioned here might be caused by misunderstanding of the literature by reviewer #3. First, on the contrary to reviewer #3’s statement that the fly-to-fly, generation-to-generation variation in rDNA copy number is minimal, Lyckegaard and Clark have demonstrated striking 6-fold variations in rDNA copy number among *D. melanogaster* populations (Lyckegaard and Clark, 1989, Lyckegaard and Clark, 1991). Also, an earlier study showed similar variation in rDNA copy number (80-600 copies) (Mohan and Ritossa, 1970), where flies with <~130 copies exhibiting bobbed phenotype (meaning that the copy number variation between 130-600, ~4.5-fold variation, is asymptomatic). Furthermore, rDNA copy number fluctuation is a broadly observed phenomenon as mentioned above, arguing that our discovery is rather in line with the existing literature.

We suspect that reviewer #3’s perception of ‘robust stability of rDNA’ comes from the fact that ‘normal’ fly populations or aging flies do not exhibit bobbed phenotype. However, the earlier studies have clearly demonstrated that asymptomatic copy number variation is fairly common. We would very much appreciate if reviewer #3 could point to any literature that refuted these studies we are referring to (if s/he is aware of any).

Second, the notion that ‘all rDNA genes, if not inserted, are currently thought to be equivalently regulated’ is not supported by the body of existing literature. Although the higher rate of transposon insertion on X chromosome was speculated to be a cause of nucleolar dominance in *Drosophila* (Greil and Ahmad, 2012), it has not been experimentally proven (which would be extremely difficult, because one would have to remove all/many transposon insertions from X chromosomes in an isogenic background to test this hypothesis). Moreover, and more importantly, the field of ‘nucleolar dominance’ is the research area to specifically investigate how particular rDNA loci are selectively activated, whereas other rDNA loci are selectively inactivated as a means of dosage control of rRNA expression (reviewed in (McStay and Grummt, 2008, Tucker et al., 2010)). This is often described as ‘a large scale epigenetic regulation, only second to X inactivation in mammalian females’(Pontes et al., 2007). Therefore, we do not believe that our results or claims are inconsistent with the existing literature, requiring reconciliation as reviewer #3 indicates.

However, the authors do not demonstrate how the observed germline changes relate to somatic rDNA levels and phenotype. They assume that the changes they observe in germ cell rDNA copy number also affect somatic cell rDNA copy number and could in an extreme case generate a defective NO. Since somatic cells could compensate significantly for the observed low level rDNA copy number variation by differential replication, and since no mutant chromosome derivatives or bobbed flies have been recovered, this is unjustified. Consequently, the final version of the paper should focus on the observed changes in germline rDNA copy number and nucleolar usage. How cyclic age dependent losses and recovery affect the soma and directional rDNA evolution, if it does so at all, remains unclear. As a result, it is premature to speculate so extensively on how these small germline rDNA copy number changes might relate to rDNA magnification. Such speculation should be cut back to 1 or two sentences and saved for a publication in which the copy number and phenotypic expression of rDNA in somatic cells is experimentally addressed.

As we have discussed in the previous round of revision, we fully agree with these comments. The inclusion of more extensive discussion on rDNA copy number in somatic cells was in response to the specific instruction requested by reviewers in the previous round. (previous review comment ‘The authors do not address the very interesting question of whether the nucleolar phenomena and rDNA copy number changes they observed are confined to germ cells. Is there nucleolar dominance, loss of genes with aging, recovery in F_1_ adults in somatic hub cells and cyst stem cells, like in GSCs? In particular, does the reduced level of Y rDNA inherited from aged fathers persist in the soma of F_1_ animals, even though rDNA copy number is restored in the germline?’). Please also note that, as we have stated in our previous response letter, we considered rDNA copy number dynamics in somatic cells to be beyond the scope of the current study (although very interesting), for the exact same reason that the reviewers felt here: thus, we limited our discussion on this issue to within our response letter and did not include any discussion on somatic cells in the main text of the previous version. Thus, we believe there is nothing to be ‘cut back’.

In addition, the line-breaks in the graphs should be removed as the authors agreed in their response.

We have revised the figures according to this suggestion.

Second, the authors should acknowledge that the FISH experiments have not been shown to be fully quantitative.

We have added this description to the revised text, which now reads as “These results suggest that our DNA FISH method is sensitive enough to distinguish differences in the relative copy number of X and Y chromosome rDNA loci between different conditions, although it might not be fully quantitative.” (subsection “Y chromosome rDNA copy number decreases in the male germline during aging”)

Reviewer #1:In the revised version and the response to reviewers, the authors have cogently addressed my concerns regarding some weaknesses and reviewer misunderstandings in the first version of this paper. This is a highly original work that addresses a central but neglected topic, namely the role of rDNA instability in stem cells during aging. The authors developed cytologically based methods that allow X and Y nucleolar activity to be visualized independently in stem cell. The in-situ hybridization data provide convincing corroboration of the PCR data that rDNA gene copy number is changing during aging and undergoing restoration in you F_1_ males. This paper will stimulate research in this field to answer many remaining questions concerning the developmental timing and mechanism of rDNA restoration, and on the epigenetic mechanisms that program rRNA gene activation and repression.

We very much appreciate this reviewer for his/her positive comments. As concisely summarized by this reviewer, we believe that the combination of cytological method (DNA FISH) and PCR provides strong evidence for rDNA copy number fluctuation. Whereas our method might not provide the resolution of rDNA copy number down to the absolute exact number, we believe that the resolution is sufficient to make our points of age-related decline and subsequent recovery.

Reviewer #2:The revised manuscript "Transgenerational dynamics of rDNA copy number in Drosophila male germline stem cells" extends analysis of changes in rDNA expression using SNPs in transcripts, and includes new data using FISH to determine gene copy number on deletion Y chromosomes. These data do help document changes in gene copy number but there are some details that should be included. The authors use two deletion Y chromosomes to demonstrate that FISH is sensitive to copy number differences. However, copy numbers of these chromosomes (determined by some other method) should be reported.

We appreciate this reviewer for his/her support.

For copy number estimation of two Ybb alleles, we do not believe that any technically feasible experiments would provide meaningful answers. The most precise estimation may be achieved by qPCR: however, to obtain rDNA copy number on particular chromosome, we would have to place it in genetic background that contains no rDNA on the other chromosome. In this case, placing Ybb in the background of Xbb. This leads to lethality for both of Df(YS)bb and Df(YS)bb- strains used in our study, thus we cannot perform such qPCR. However, for Df(YS)bb (partial deletion), based on Y:X rDNA ratio shown in Figure 3, we can estimate this Ybb allele contains 15-32% of wild type copy number. For Df(YS)bb-, its rDNA is under the detection limit by our rDNA FISH method (Figure 3—figure supplement 1). We believe that this level of resolution provides sufficient information for the purpose of our current study. We also stated that our method may not be fully quantitative.

The authors seem to be using two different deletion Y chromosomes, but the names of these chromosomes are confusing (Df(YS)bb is different from Df(YS)bb-?).

We fully agree and had sought for the way to name them more clearly (we had a lengthy discussion among authors), such as bb^partial^ vs. bb^complete^. However, Df(YS)bb- and Df(YS)bb are the terms used by the original authors who described these deficiencies and listed as such in FlyBase. We concluded that, if we rename, it would cause further confusion for the future readers who try to track down the information, corresponding our “bb^partial^” to “bb^complete^”. Therefore, we decided to cite the original papers that described each allele and adhere to original nomenclature.

A central issue that was raised in the first review was whether aging throws bobbed-deficient Y chromosomes, that should be apparent by some frequency of bobbed phenotypes in progeny. The authors allude that they have some data on this, but this is not clearly described. They should include a more thorough description and data supporting the statement in subsection “rDNA copy number maintenance through generations” that "we barely observed bobbed phenotype even among the sons of very old fathers". Given the extent of copy number loss the authors are estimating, such bobbed progeny are expected.

As we discussed in the first round of revision, it was described that bobbed phenotype arises when rDNA copy number is lower than 130 in total (Ritossa et al., 1966), although this number may vary depending on genetic background. This indicates that rDNA copy number can fluctuate without showing bobbed phenotype, which is consistent with earlier work that showed copy number variation of 130-600 while being asymptomatic (Mohan and Ritossa, 1970). As the Figure 3 suggests, the reduction in Y rDNA is around ~50% in aged males, the level that would not necessarily show bb phenotype even when combined with Xbb chromosome (rDNA complete loss). Indeed, when 40 day old fathers are mated with bb^158^/FM7 females, we obtained 428 male bb^158^/Y off spring and 424 female bb158/X offspring, and none of them were bobbed. This indicates that rDNA copy number decrease is not sufficient to exhibit bobbed phenotype even when placed in the background of Xbb (bb^158^).

Reviewer #3:This is a re-review of "Transgenerational dynamics of rDNA copy number in Drosophila male germline stem cells,", submitted by Yukiko Yamashita and colleagues.

We thank this reviewer for thoroughly reviewing our manuscript. However, we’re afraid that some of concerns raised by this reviewer are ungrounded. As detailed below, our observations are consistent with existing literature and we do not find a need of ‘reconciliation’.

My concerns remain, even after the authors sought to clarify their findings.First, I remain very concerned that the vast literature on rDNA magnification remains unaddressed, even though those data are at odds with the observations made here. That concern is more profound because the measurements of rDNA loss here are not direct and simple: the authors use nucleolar shape, Y:X ratio, unvalidated FISH, and qPCR quantification on one tissue relatively late in development (eclosed adult testes) as proxy for rDNA copy number in sperm; more direct and more reliable assays are available.

In fact, our data are NOT at odds with the existing literature. This reviewer states that rDNA copy number is stably maintained with fly-to-fly, generation-to-generation variations being minimal. On the contrary to this statement, a large body of literature has described variations in rDNA copy number (Lyckegaard and Clark, 1989, Lyckegaard and Clark, 1991, Mohan and Ritossa, 1970) with striking 6-fold difference without showing detectable bobbed phenotype. Although these studies did not specifically consider aging as a parameter that may affect rDNA copy number (which is one of our major discovery in this study), their study unlikely controlled for the age of flies, and thus we suspect that a part of the copy number variation they detected may reflect aging.

We suspect that this reviewer assumed the fact that there is little fly-to-fly, generation-to-generation variation in the frequency of bobbed flies as indication of little copy number variation. However, previous studies and our study agree that copy number variation is not necessarily symptomatic, as bobbed phenotype only appears when copy number is extremely low (less than 130 copies, as estimated by (Ritossa et al., 1966)).

As we already described above, rDNA copy number variation is not unique to *Drosophila*, and broadly observed in many species including humans.

The authors assert that rDNA counts drop with aging males, that those lower counts are still detectable (and are to the same degree) in testes of newly-eclosed males, they recover soon, and are lost again (presumably due to aging). The essential experiment that was called for in the first review was to measure loss in the whole soma of the progeny (in some type of rDNA[0]/Y flies – males of sc[4]sc[8]Y or females of DX,rDNA[0]/Y). The authors state that they did not observe any bobbed phenotype, which is perhaps expected if the original Y had a large number of rDNA copies, however losses should be easily detectable in whole animals using qPCR (as Aldrich and Maggert did), or the authors could start with shorter rDNA array containing Ys, such as one of the bobbed alleles to which they have access, and measure loss based on bobbed phenotype. The justification for asking this is simple: the Ys from old males should have reduced rDNA copy number, obvious in all of the soma of the offspring. In the case of the bobbed literature, this would have been noticed (by > dozens of reports) as an increase in the bobbed expressivity or penetrance, which was never reported; in fact rDNA copy number is remarkably robust fly-to-fly and generation-to-generation, which is at odds with the observations here. Either those previous studies are wrong, or they did not do the crosses in a way to detect such a remarkable variation in rDNA copy number, or the observations here are not detecting rDNA loss. Whichever the answer is, this has to be resolved prior to accepting the author's conclusion of their data (Points 1, 3, 10).

In the previous round of revision, we have provided explanations for somatic cells’ rDNA copy number is not relevant to our discovery described here, to which the majority of the reviewers seemed to have agreed and suggested not to discuss about somatic rDNA copy number (some more details are provided at the beginning of this response).

As an aside, in terms of rDNA copy number changes, nothing in the authors' data rule out that they are observing a process of endoreduplication/polytenization in a subset of cells in the testes (as they allude to in their response to Point 5). If that is the case, the apparent "losses" and "gains" would not bear on the sperm haploid genomes, which would be consistent with both the authors' measured effects and the known genetic stability from others' work on bobbed magnification. Without looking in the soma of the offspring, they cannot know. This experiment, in my mind, remains essential.

It is in theory possible that somatic gonadal cells undergo endoreplication and polytenization, influencing our qPCR results. However, if it were the case, we would not expect to see changes in rDNA copy number in F1s from old father (obviously, somatic gonadal cells will not be transmitted to the next generation, polytenized or not).

Second, I am unconvinced by the authors' assurance that their FISH is quantitative. Their care to not saturate the signal is fine, but they do not show any data that it is quantitative (which would require determining a lowest-threshold, a linearity of response between that threshold and saturation, and ample controls/experiments showing that photobleaching, preferential binding, etc. are not issues). I think that a lower threshold must exist since the authors cannot detect their hypothetical extrachromosomal rDNA (Lines 371-374). FISH is not intrinsically quantitative and cannot be used as such unless great pains are taken. Hence, treating or calling it quantitative is not appropriate (Point 1).

According to the suggestion by reviewers, we explained that our method might not be fully quantitative (see above). We also noted that photobleaching is minimal (no detectable bleaching during our imaging). As we had explained in the previous round of revision that there would not be ‘preferential binding’ of probes, because DNA FISH was conducted with Stellaris tiled probes (48 oligos) that do not distinguish SNPs (detailed in method, and probe sequences had been provided in the Supplementary file 3). Thus, it is highly unlikely there will be preferential binding between X vs Y rDNA locus.

*Related to this, I'd still like to know how exclusive* and *abundant these SNPs are to the two (X-linked and Y-linked) rDNA arrays (Point 8).*

Figure 2—figure supplement 1 had been provided for this purpose. Y SNP signal is undetectable with our method in XO flies, and X SNP signal is undetectable in Xbb/Y flies, suggesting that SNPs are quite specific (if not ‘exclusive’) to individual locus. As was described in the method, DNA sequencing to determine SNPs were conducted using genomic DNA from XO and Xbb/Y without cloning a single rDNA copy from them. Therefore, if various SNPs exist as a mixture among rDNA copies within a rDNA locus, we would not have been able to ‘read’ the sequence well (chromatogram showing peaks of mixed signals). Therefore, it is fairly reasonable to assume the SNPs are highly homogenous and robust within individual rDNA locus. This is consistent with the well-established notion of ‘concerted evolution’ of repetitive sequence, which homogenizes mutations within the repeats across the locus (reviewed in (Eickbush and Eickbush, 2007)).

Third, I still do not understand how the overall structure of the Y-linked array is consistent with the authors' model. If R1- and R2- inserted rDNA copies are interspersed with uninserted rDNA, I cannot imagine a damage-and-repair mechanism (that involves intrachromosomal HR-based recombination) that does not remove intervening rDNA. Either way, the structure of the Ys (i.e., degree of interspersion/clustering of R1- and R2- elements in the entire rDNA gene cluster) being used in this study should be included, as we asked (Points 8 and 12).

This issue has been beautifully explained by (Zhou et al., 2013) using modeling, and our results are consistent with their model. Most importantly, however, the present study is simply reporting the experimental data that uninserted copies seem to be specifically lost, which is not affected by the presence of, but happens to fit well with, the existing modeling: thus, it is not reasonable to argue that our experimental data cannot be true, because one cannot imagine how it can happen.

In general, I am not satisfied with the discussion of magnification and how that bears on these studies. I am not convinced they are the same thing, and I do not understand how the authors envision their proposed phenomenon and magnification are related. In subsection “rDNA copy number maintenance through generations”, they propose unequal sister exchange as a mechanism, however the authors should explain the fate of the other (shortened) product of such an exchange. Are they detectable in their data? (Points 3, 11).

We would like to point out that no studies to date have shown the fate of shortened copy after unequal sister chromatid exchange. (Tartof, 1974) noted the emergence of a small number of individuals with worsened bobbed phenotype, while the flies undergo magnification, leading Tartof to speculate they might represent the individuals who inherited ‘shortened’ copies. We acknowledge that we do not fully understand how rDNA copy number recovers in young flies from old fathers: however, it is also true that no evidence has been provided as to exactly how magnification happens, either, and nobody knows where the shorted copies may go during unequal sister chromatid exchange that happens during magnification. Even more critically, although gene requirements (mus^-1^01 etc.) and other circumstantial evidence have pointed to unequal sister chromatid exchange as a likely mechanism for magnification, this is not directly proven yet. In fact, (de Cicco and Glover, 1983) concluded that ‘unequal sister chromatid exchange cannot explain magnification, unless a single cell can undergo such a process repeatedly’. In our discussion in the manuscript, we mentioned that it might be explained if GSCs (which can undergo repeated round of asymmetric division) are the place for magnification, with clear indication that this is a pure speculation. As is clear by now, the request of examining the fate of shortened copy as a proof of our model clearly goes beyond the scope of this study.

I continue to be uncomfortable with the data presentation, with line breaks accentuating the magnitude of effects. Breaks are conventionally used to graph data with vastly different values on the same graph, where here they are used consistently but have the effect of making smaller effects seem larger. Graphs should give a visual representation of the size of effects, but the use in this paper overemphasizes them (sometimes to a very large degree). This change remains essential (Point 9).

We have revised all the graphs accordingly.

The question as to whether an X, in X/Ybb males, is also subject to these losses seems central to the authors' model. It can – and should be – addressed experimentally (Point 10).

We do not believe the nature of X rDNA dynamics is central to our model. If X rDNA does not change at all, it would only mean that our data indicate a striking decrease of Y rDNA copy number, and all data presented in this study becomes more quantitative in nature (for example, when we measure Y rDNA copy number as Y/X ratio, we would underestimate Y copy number decrease, if the denominator (X) is also decreasing. This concern will be eliminated if X is not losing copy number. If X rDNA is being lost from X during aging as well, this would mean that rDNA copy number loss is likely due to inherent nature of repeat, and it is not due to anything special to Y chromosome, or Y rDNA. Our discussion and statements are inclusive to both models, and we do not believe that distinction of these two possibilities is the core impact of our discovery.

Our explanation provided in the last round of revision was as follows:

“Indeed, X/Df(YS)bb flies would have active X rDNA, which is clearly indicated in Figure 3. And we certainly predict that X rDNA would be more destabilized. However, we would like to clarify that we did not/do not claim that silent X rDNA is entirely stable. The core message of the present study is dynamic nature of rDNA copy number (decrease during aging, recovery in the next generation), and we do not think this is X or Y chromosome-specific phenomenon. Transcriptional state accelerates but not solely determines the instability-thus, Y rDNA is more affected than X rDNA, but not exclusively. Indeed, the data in Figure 4 suggest that X chromosome is also somewhat compromised during aging, and we had discussed that X rDNA is also likely affected during aging. We have now clarified this point in the revised text not to leave the impression that rDNA instability occurs exclusively on the Y chromosome.

Regarding the nature of the Df(YS)bb chromosome, it is the Y rDNA deficiency stock listed at the Bloomington *Drosophila* Stock Center, and is previously characterized having insufficient Y rDNA for viability when X rDNA is disrupted (Cline 2001). This reference is now included in the text. The nature of ‘partial deletion’ of this chromosome is described in this study by DNA FISH, which shows clear reduction (but not complete deletion) of Y rDNA (Figure 3).”

We hope that this makes it clear that knowing whether X rDNA may undergo copy number reduction or not does not impact the value of our present study.

These above experiments were originally denoted as "essential," but they were not done in this revision. It is my opinion that this study – if correct – could overturn decades of research on rDNA copy number determination, rDNA stability and changes, and the work on rDNA magnification. It is thus critical that the appropriate experiments be done to rule out alternatives and to pursue the specific predictions of these proposed models.

As we hope is clear by now (based on explanation provided above), our results are not overturning decades of research: on the contrary to this perception, our results are in fact in line with existing knowledge, while providing advancement.